# Learning Temporal Coherence via Self-Supervision for GAN-based Video Generation

## Abstract

We focus on temporal self-supervision for GAN-based video generation tasks. While adversarial training successfully yields generative models for a variety of areas, temporal relationship in the generated data is much less explored. This is crucial for sequential generation tasks , e.g. video super-resolution and unpaired video translation. For the former, state-of-the-art methods often favor simpler norm losses such as $L^2$ over adversarial training. However, their averaging nature easily leads to temporally smooth results with an undesirable lack of spatial detail. For unpaired video translation, existing approaches modify the generator networks to form spatio-temporal cycle consistencies. In contrast, we focus on improving the learning objectives, and propose a temporally self-supervised algorithm. For both tasks, we show that temporal adversarial learning is key to achieving temporally coherent solutions without sacrificing spatial detail. We also propose a novel Ping-Pong loss to improve the long-term temporal consistency. It effectively prevents recurrent networks from accumulating artifacts temporally without depressing detailed features. We also propose a first set of metrics to quantitatively evaluate the accuracy as well as the perceptual quality of the temporal evolution. A series of user studies confirms the rankings computed with these metrics.

## 1 Introduction

Generative adversarial models (GANs) have been extremely successful at learning complex distributions such as natural images (Zhu et al., 2017; Isola et al., 2017). However, for sequence generation, directly applying GANs without carefully engineered constraints typically results in strong artifacts over time due to the significant difficulties introduced by the temporal changes. In particular, conditional video generation tasks are very challenging learning problems where generators should not only learn to represent the data distribution of the target domain, but also learn to correlate the output distribution over time with conditional inputs. Their central objective is to faithfully reproduce the temporal dynamics of the target domain and not resort to trivial solutions such as features that arbitrarily appear and disappear over time.

In our work, we propose a novel adversarial learning method for a recurrent training approach that supervises both spatial content as well as temporal relationships. We apply our approach to two video-related tasks that offer substantially different challenges: *video super-resolution* (VSR) and *unpaired video translation* (UVT). With no ground truth motion available, the spatio-temporal adversarial loss and the recurrent structure enable our model to generate realistic results while keeping the generated structures coherent over time. With the two learning tasks we demonstrate how spatio-temporal adversarial training can be employed in paired as well as unpaired data domains. In addition to the adversarial network which supervises the short-term temporal coherence, long-term consistency is self-supervised using a novel bi-directional loss formulation, which we refer to as "Ping-Pong" (PP) loss in the following. The PP loss effectively avoids the temporal accumulation of artifacts, which can potentially benefit a variety of recurrent architectures. The central contributions of our work are: a spatio-temporal discriminator unit together with a careful analysis of training objectives for realistic and coherent video generation tasks, a novel PP loss supervising long-term consistency, in addition to a set of metrics for quantifying temporal coherence based on motion estimation and perceptual distance. Together, our contributions lead to models that outperform previous work in terms of temporally-coherent detail, which we quantify with a wide range of metrics and user studies.

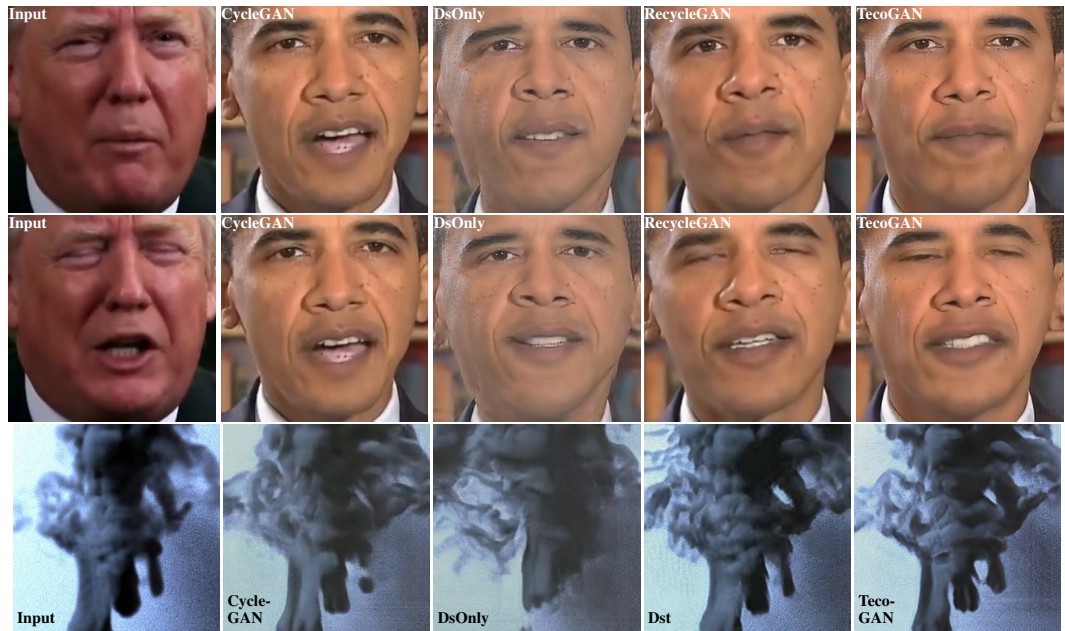

Figure 1: When learning a mapping between Trump and Obama, the CycleGAN model gives good spatial features, but collapses to essentially static outputs of Obama. It manages to transfer facial expressions back to Trump using tiny differences encoded in its Obama outputs, instead of learning a meaningful mapping. Being able to establish the correct temporal cycle-consistency between domains, ours and RecycleGAN can generate correct blinking motions. Our model outperforms the latter in terms of coherent detail that is generated.

## 2 RELATED WORK

Deep learning has made great progress for image generation tasks. While regular losses such as $L^2$ (Kim et al., 2016; Lai et al., 2017) offer good performance for image super-resolution (SR) tasks in terms of PSNR metrics, GAN researchers found adversarial training (Goodfellow et al., 2014) to significantly improve the perceptual quality in multi-modal problems including image SR (Ledig et al., 2016), image translations (Zhu et al., 2017; Isola et al., 2017), and others. Perceptual metrics (Zhang et al., 2018; Prashnani et al., 2018) are proposed to reliably evaluate image similarity by considering semantic features instead of pixel-wise errors.

Video generation tasks, on the other hand, require realistic results to change naturally over time. Recent works in VSR improve the spatial detail and temporal coherence by either using multiple low-resolution (LR) frames as inputs (Jo et al., 2018; Tao et al., 2017; Liu et al., 2017), or recurrently using previously estimated outputs (Sajjadi et al., 2018). The latter has the advantage to re-use high-frequency details over time. In general, adversarial learning is less explored for VSR and applying it in conjunction with a recurrent structure gives rise to a special form of temporal mode collapse, as we will explain below. For video translation tasks, GANs are more commonly used but discriminators typically only supervise the spatial content. E.g., Zhu et al. (2017) does not employ temporal constrains and generators can fail to learn the temporal cycle-consistency. In order to learn temporal dynamics, RecycleGAN (Bansal et al., 2018) proposes to use a prediction network in addition to a generator, while a concurrent work (Chen et al., 2019) chose to learn motion translation in addition to spatial content translation. Being orthogonal to these works, we propose a spatio-temporal adversarial training for both VSR and UVT and we show that temporal self-supervision is crucial for improving spatio-temporal correlations without sacrificing spatial detail. While $L^2$ temporal losses based on warping are used to enforce temporal smoothness in video style transfer tasks (Ruder et al., 2016; Chen et al., 2017), concurrent GAN-based VSR work (Pérez-Pellitero et al., 2018) and UVT work (Park et al., 2019), it leads to an undesirable smooth over spatial detail and temporal changes in outputs. Likewise, the $L^2$ temporal metric represents a sub-optimal way to quantify temporal coherence and perceptual metrics that evaluate natural temporal changes are

unavailable up to now. We work on this open issue, propose two improved temporal metric and demonstrate the advantages of temporal self-supervision over direct temporal losses.

Previous work, e.g. tempoGAN (Xie et al., 2018) and vid2vid (Wang et al., 2018b), have proposed adversarial temporal losses to achieve time consistency. While tempoGAN employs a second temporal discriminator with multiple aligned frames to assess the realism of temporal changes, it is not suitable for videos, as it relies on ground truth motions and employs a single-frame processing that is sub-optimal for natural images. On the other hand, vid2vid focuses on paired video translations and proposes a video discriminator based on a conditional motion input that is estimated from the paired ground-truth sequences. We focus on more difficult unpaired translation tasks instead, and demonstrate the gains in quality of our approach in the evaluation section. For tracking and optical flow estimation, L2-based time-cycle losses (Wang et al., 2019) were proposed to constrain motions and tracked correspondences using symmetric video inputs. By optimizing indirectly via motion compensation or tracking, this loss improves the accuracy of the results. For video generation, we propose a PP loss that also makes use of symmetric sequences. However, we directly constrain the PP loss via the generated video content, which successfully improves the long-term temporal consistency in the video results.

# 3 LEARNING TEMPORALLY COHERENT CONDITIONAL VIDEO GENERATION

**Generative Network** Before explaining the temporal self-supervision in more detail, we outline the generative model to be supervised. Our generator networks produce image sequences in a frame-recurrent manner with the help of a recurrent generator G and a flow estimator $F$. We follow previous work (Sajjadi et al., 2018), where G produces output $g_t$ in the target domain B from conditional input frame $a_t$ from the input domain A, and recursively uses the previous generated output $g_{t-1}$. $F$ is trained to estimate the motion $v_t$ between $a_{t-1}$ and $a_t$, which is then used as a motion compensation that aligns $g_{t-1}$ to the current frame. This procedure, also shown in Fig. 2a), can be summarized as: $g_t = G(a_t, W(g_{t-1}, v_t))$, where $v_t = F(a_{t-1}, a_t)$ and $W$ is the warping operation. While one generator is enough to map data from A to B for

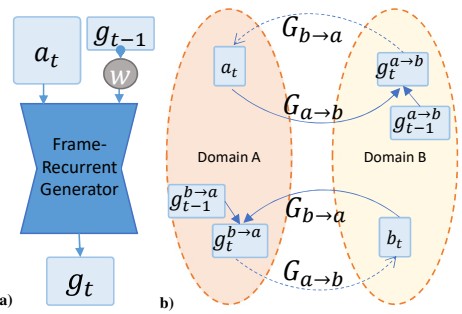

Figure 2: a) G. b) The UVT cycle link using recurrent G.

paired tasks such as VSR, unpaired generation requires a second generator to establish cycle consistency. (Zhu et al., 2017). In the UVT task, we use two recurrent generators, mapping from domain A to B and back. As shown in Fig. 2b), given $g_t^{a \to b} = G_{ab}(a_t, W(g_{t-1}^{a \to b}, v_t))$, we can use $a_t$ as the labeled data of $g_t^{a \to b \to a} = G_{ba}(g_t^{a \to b}, W(g_{t-1}^{a \to b \to a}, v_t))$ to enforce consistency. A ResNet architecture is used for the VSR generator G and a encoder-decoder structure is applied to UVT generators and $F$. We intentionally keep generators simple and in line with previous work, in order to demonstrate the advantages of the temporal self-supervision that we will explain in the following paragraphs.

**Spatio-Temporal Adversarial Self-Supervision** The central building block of our approach is a novel *spatio-temporal* discriminator $D_{s,t}$ that receives triplets of frames. This contrasts with typically used *spatial* discriminators which supervise only a single image. By concatenating multiple adjacent frames along the channel dimension, the frame triplets form an important building block for learning because they can provide networks with gradient information regarding the realism of spatial structures as well as short-term temporal information, such as first- and second-order time derivatives.

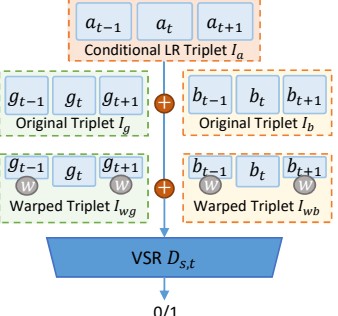

Figure 3: Conditional VSR $D_{s,t}$.

We propose a $D_{s,t}$ architecture, illustrated in Fig. 3 and Fig. 4, that primarily receives two types of triplets: three adjacent frames and the corresponding warped ones. We warp later frames backward and previous ones forward. While original frames contain the full spatio-temporal infor-

mation, warped frames more easily yield temporal information with their aligned content. For the input variants we use the following notation: $I_g = \{g_{t-1}, g_t, g_{t+1}\}$, $I_b = \{b_{t-1}, b_t, b_{t+1}\}$; $I_{wg} = \{W(g_{t-1}, v_t), g_t, W(g_{t+1}, v'_t)\}$, $I_{wb} = \{W(b_{t-1}, v_t), b_t, W(b_{t+1}, v'_t)\}$.

For VSR tasks, $D_{s,t}$ should guide the generator to learn the correlation between LR inputs and high-resolution (HR) targets. Therefore, three LR frames $I_a = \{a_{t-1}, a_t, a_{t+1}\}$ from the input domain are used as a conditional input. The input of $D_{s,t}$ can be summarized as $I^b_{s,t} = \{I_b, I_{wb}, I_a\}$ labelled as *real* and the generated inputs $I^g_{s,t} = \{I_g, I_{wg}, I_a\}$ labelled as *fake*. In this way, the conditional $D_{s,t}$ will penalize $G$ if $I_g$ contains less spatial details or unrealistic artifacts according to $I_a, I_b$. At the same time, temporal relationships between the generated images $I_{wg}$ and those of the ground truth $I_{wb}$ should match. With our setup, the discriminator profits from the warped frames to classify realistic and unnatural temporal changes, and for situations where the motion estimation is less accurate, the discriminator can fall back to the original, i.e. not warped, images.

For UVT tasks, we demonstrate that the temporal cycle-consistency between different domains can be established using the supervision of unconditional spatio-temporal discriminators. This is in contrast to previous work which focuses on the generative networks to form spatio-temporal cycle links. Our approach actually yields improved results, as we will show below, and Fig. 1 shows a preview of the quality that can be achieved using spatio-temporal discriminators. In practice, we found it crucial to ensure that generators first learn reasonable spatial features, and only then improve their temporal correlation. Therefore, different to the $D_{s,t}$ of VST that always receives 3 concatenated triplets as an input, the unconditional $D_{s,t}$ of UVT only takes one triplet at a time. Focusing on the generated data, the input for a single

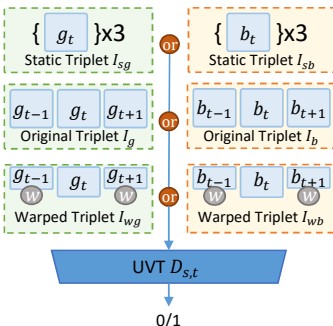

Figure 4: Unconditional UVT $D_{s,t}$.

batch can either be a static triplet of $I_{sg} = \{g_t, g_t, g_t\}$, the warped triplet $I_{wg}$, or the original triplet $I_g$. The same holds for the reference data of the target domain, as shown in Fig. 4. With sufficient but complex information contained in these triplets, transition techniques are applied so that the network can consider the spatio-temporal information step by step, i.e., we initially start with 100% static triplets $I_{sg}$ as the input. Then, over the course of training, 25% of them transition to $I_{wg}$ triplets with simpler temporal information, with another 25% transition to $I_g$ afterwards, leading to a (50%,25%,25%) distribution of triplets. Details of the transition calculations are given in Appendix D. Here, the warping is again performed via $F$.

While non-adversarial training typically employs loss formulations with static goals, the GAN training yields dynamic goals due to discriminative networks discovering the learning objectives over the course of the training run. Therefore, their inputs have strong influence on the training process and the final results. Modifying the inputs in a controlled manner can lead to different results and substantial improvements if done correctly, as will be shown in Sec. 4. Although the proposed concatenation of several frames seems like a simple change that has been used in a variety of projects, it is an important operation that allows discriminators to understand spatio-temporal data distributions. As will be shown below, it can effectively reduce temporal problems encountered by spatial GANs. While $L^2-$based temporal losses are widely used in the field of video generation, the spatio-temporal adversarial loss is crucial for preventing the inference of blurred structures in multi-modal data-sets. Compared to GANs using multiple discriminators, the single $D_{s,t}$ network can learn to balance the spatial and temporal aspects from the reference data and avoid inconsistent sharpness as well as overly smooth results. Additionally, by extracting shared spatio-temporal features, it allows for smaller network sizes.

**Self-Supervision for Long-term Temporal Consistency** When relying on a previous output as input, i.e., for frame-recurrent architectures, generated structures easily accumulate frame by frame. In an adversarial training, generators learn to heavily rely on previously generated frames and can easily converge towards strongly reinforcing spatial features over longer periods of time. For videos, this especially occurs along directions of motion, and these solutions can be seen as a special form of temporal mode collapse. We have noticed this issue in a variety of recurrent architectures, examples are shown in Fig. 5 a) and the Dst in Fig. 1. While this issue could be alleviated by training with longer sequences, we generally want generators to be able to work with sequences of arbitrary length for inference. To address this inherent problem of recurrent generators, we propose a new

Figure 5: a) Result without PP loss. The VSR network is trained with a recurrent frame-length of 10. When inference on long sequences, frame 15 and latter frames of the foliage scene show the drifting artifacts. b) Result trained with PP loss. These artifacts are removed successfully for the latter. c) The ground-truth image. With our PP loss (shown on the right), the $L^2$ distance between $g_t$ and $g_t'$ is minimized to remove drifting artifacts and improve temporal coherence.

bi-directional "Ping-Pong" loss. For natural videos, a sequence with forward order as well as its reversed counterpart offer valid information. Thus, from any input of length $n$, we can construct a symmetric PP sequence in form of $a_1, ...a_{n-1}, a_n, a_{n-1}, ...a_1$ as shown in Fig. 5. When inferring this in a frame-recurrent manner, the generated result should not strengthen any invalid features from frame to frame. Rather, the result should stay close to valid information and be symmetric, i.e., the forward result $g_t = G(a_t, g_{t-1})$ and the one generated from the reversed part, $g_t' = G(a_t, g_{t+1}')$, should be identical. Based on this observation, we train our networks with extended PP sequences and constrain the generated outputs from both "legs" to be the same using the loss: $\mathcal{L}_{pp} = \sum_{i=1}^{n-1} \|g_t - g_t'\|_2$. Note that in contrast to the generator loss, the $L^2$ norm is a correct choice here: We are not faced with multi-modal data where an $L^2$ norm would lead to undesirable averaging, but rather aim to constrain the recurrent generator to its own, unique version over time. The PP terms provide constraints for short term consistency via $\|g_{n-1} - g_{n-1}'\|_2$, while terms such as $\|g_1 - g_1'\|_2$ prevent long-term drifts of the results. As shown in Fig. 5(b), this PP loss successfully removes drifting artifacts while appropriate high-frequency details are preserved. In addition, it effectively extends the training data set, and as such represents a useful form of data augmentation. A comparison is shown in Appendix E to disentangle the effects of the augmentation of PP sequences and the temporal constrains. The results show that the temporal constraint is the key to reliably suppressing the temporal accumulation of artifacts, achieving consistency, and allowing models to infer much longer sequences than seen during training.

**Perceptual Loss Terms** As perceptual metrics, both pre-trained NNs (Johnson et al., 2016; Wang et al., 2018a) and in-training discriminators (Xie et al., 2018) were successfully used in previous work. Here, we use feature maps from a pre-trained VGG-19 network (Simonyan & Zisserman, 2014), as well as $D_{s,t}$ itself. In the VSR task, we can encourage the generator to produce features similar to the ground truth ones by increasing the cosine similarity between their feature maps. In UVT tasks without paired ground truth data, we still want the generators to match the distribution of features in the target domain. Similar to a style loss in traditional style transfer (Johnson et al., 2016), we here compute the $D_{s,t}$ feature correlations measured by the Gram matrix instead. The feature maps of $D_{s,t}$ contain both spatial and temporal information, and hence are especially well suited for the perceptual loss.

**Loss and Training Summary** We now explain how to integrate the spatio-temporal discriminator into the paired and unpaired tasks. We use a standard discriminator loss for the $D_{s,t}$ of VSR and a least-square discriminator loss for the $D_{s,t}$ of UVT. Correspondingly, a non-saturated $\mathcal{L}_{adv}$ is used for the $G$ and $F$ of VSR, and a least-squares one is used for the UVT generators. As summarized in Table 1, $G$ and $F$ are trained with the mean squared loss $\mathcal{L}_{\text{content}}$, adversarial losses $\mathcal{L}_{adv}$, perceptual losses $\mathcal{L}_\phi$, the PP loss $\mathcal{L}_{\text{PP}}$, and a warping loss $\mathcal{L}_{\text{warp}}$, where again $g$, $b$ and $\Phi$ stand for generated samples, ground truth images and feature maps of VGG-19 or $D_{s,t}$. We only show losses for the mapping from A to B for UVT tasks, as the backward mapping simply mirrors the terms. We refer to our full model for both tasks as *TecoGAN* below.[1] Training parameters and details are given in Appendix G.

---

[1] Source code, training data, and trained models will be published upon acceptance.

Table 1: Summary of loss terms.

| Loss for | VSR, $D_{s,t}$ | UVT, $D_{s,t}^b$ |
|---|---|---|
| $\mathcal{L}_{D_{s,t}}$ | $-\mathbb{E}_{b\sim p_{\mathrm{b}}(b)}[\log D(\mathrm{I}_{s,t}^b)] - \mathbb{E}_{a\sim p_{\mathrm{a}}(a)}[\log(1 - D(\mathrm{I}_{s,t}^g))]$ | $\mathbb{E}_{b\sim p(b)}[D(\mathrm{I}_{s,t}^b) - 1]^2 + \mathbb{E}_{a\sim p(a)}[D(\mathrm{I}_{s,t}^g)]^2$ |

| Loss for | VSR, G & $F$ | UVT, $G_{ab}$ |
|---|---|---|
| $\mathcal{L}_{G,F}$ | $\lambda_c \mathcal{L}_{\mathrm{content}} + \lambda_a \mathcal{L}_{\mathrm{adv}} + \lambda_\phi \mathcal{L}_\phi + \lambda_p \mathcal{L}_{\mathrm{PP}} + \lambda_w \mathcal{L}_{\mathrm{warp}}$ | |
| $\mathcal{L}_{\mathrm{content}}$ | $\|g_t - b_t\|_2$ | $\left\|g_t^{a\to b\to a} - a_t\right\|_2 + \left\|g_t^{b\to a\to b} - b_t\right\|_2$ |
| $\mathcal{L}_{\mathrm{adv}}$ | $-\mathbb{E}_{a\sim p_{\mathrm{a}}(a)}[\log D_{s,t}(\mathrm{I}_{s,t}^g)]$ | $-\mathbb{E}_{a\sim p_{\mathrm{a}}(a)}[D_{s,t}^b(\mathrm{I}_{s,t}^{g_{a\to b}})]^2$ |
| $\mathcal{L}_\phi$ | 1.0 - $\Phi(\mathrm{I}_{s,t}^g) * \Phi(\mathrm{I}_{s,t}^b) / \left\|\Phi(\mathrm{I}_{s,t}^g)\right\| * \left\|\Phi(\mathrm{I}_{s,t}^b)\right\|$ | $\left\|GM(\Phi(\mathrm{I}_{s,t}^g)) - GM(\Phi(\mathrm{I}_{s,t}^b))\right\|_2$ |
| $\mathcal{L}_{\mathrm{PP}}$ | $\sum_{i=1}^{n-1} \|g_t - g_t'\|_2$ | |
| $\mathcal{L}_{\mathrm{warp}}$ | $\sum \|a_t - W(a_{t-1}, \mathrm{F}(a_{t-1}, a_t))\|_2$ | 0.0, a pre-trained F is used |

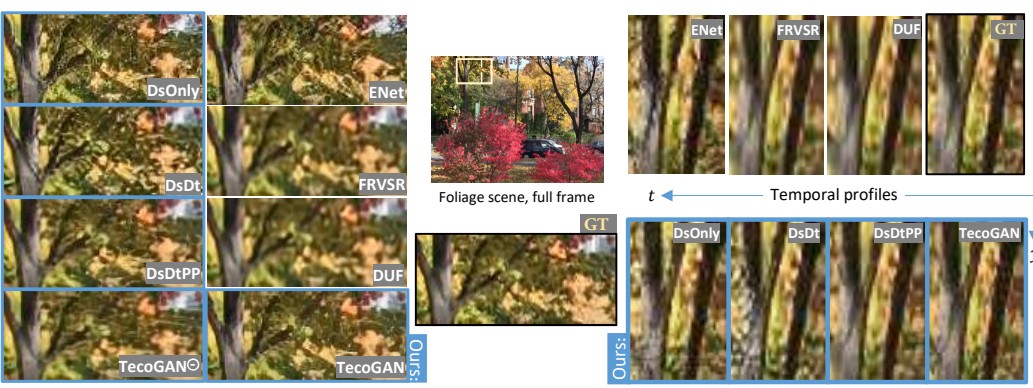

Figure 6: In VSR of the foliage scene, adversarial models (ENet, DsOnly, DsDt, DsDtPP, TecoGAN$^\odot$ and TecoGAN) yield better perceptual quality than methods using $L^2$ loss (FRVSR and DUF). In temporal profiles on the right, DsDt, DsDtPP and TecoGAN show significantly less temporal discontinuities compared to ENet and DsOnly. The temporal information of our discriminators successfully suppresses these artifacts.

## 4    ANALYSIS AND EVALUATION OF LEARNING OBJECTIVES

In the following, we illustrate the effects of temporal supervision using two ablation studies. In the first one, models trained with ablated loss functions show how $\mathcal{L}_{\mathrm{adv}}$ and $\mathcal{L}_{\mathrm{PP}}$ change the overall learning objectives. Next, full UVT models are trained with different $D_{s,t}$ inputs. This highlights how differently the corresponding discriminators converge to different spatio-temporal equilibriums, and the general importance of providing suitable data distributions from the target domain. While we provide qualitative and quantitative evaluations in the following, we also refer the reader to our supplemental *html document* [2], with video clips that more clearly highlight the temporal differences.

**Loss Ablation Study**   Below we compare variants of our full TecoGAN model to EnhanceNet (ENet) (Sajjadi et al., 2017), FRVSR (Sajjadi et al., 2018), and DUF (Jo et al., 2018) for VSR, and CycleGAN (Zhu et al., 2017) and RecycleGAN (Bansal et al., 2018) for UVT. Specifically, ENet and CycleGAN represent state-of-the-art single-image adversarial models without temporal information, FRVSR and DUF are state-of-the-art VSR methods without adversarial losses, and RecycleGAN is a spatial adversarial model with a prediction network learning the temporal evolution.

For VSR, we first train a *DsOnly* model that uses a frame-recurrent $G$ and $F$ with a VGG-19 loss and only the regular spatial discriminator. Compared to ENet, which exhibits strong incoherence due to the lack of temporal information, DsOnly improves temporal coherence thanks to the frame-recurrent connection, but there are noticeable high-frequency changes between frames. The temporal profiles of DsOnly in Fig. 6 and 8, correspondingly contain sharp and broken lines. When adding a temporal discriminator in addition to the spatial one (*DsDt*), this version generates more

---

[2]Anonymized and time-stamped supplemental material availabble at:
https://www.dropbox.com/sh/n07l8n51slh1e9c/AAAVngT9xsSzs1pJQqe5xV1Oa?dl=0.

coherent results, and its temporal profiles are sharp and coherent. However, DsDt often produces the drifting artifacts discussed in Sec. 3, as the generator learns to reinforce existing details from previous frames to fool $D_s$ with sharpness, and satisfying $D_t$ with good temporal coherence in the form of persistent detail. While this strategy works for generating short sequences during training, the strengthening effect can lead to very undesirable artifacts for long-sequence inferences. By adding the self-supervision for long-term temporal consistency $\mathcal{L}_{pp}$, we arrive at the *DsDtPP* model, which effectively suppresses these drifting artifacts with an improved temporal coherence. In Fig. 6 and Fig. 8, DsDtPP results in continuous yet detailed temporal profiles without streaks from temporal drifting. Although DsDtPP generates good results, it is difficult in practice to balance the generator and the two discriminators. The results shown here were achieved only after numerous runs manually tuning the weights of the different loss terms. By using the proposed $D_{s,t}$ discriminator instead, we get a first complete model for our method, denoted as *TecoGAN$^{\ominus}$*. This network is trained with a discriminator that achieves an excellent quality with an effectively halved network size, as illustrated on the right of Fig. 7. The single discriminator correspondingly leads to a significant reduction in resource usage. Using two discriminators requires ca. 70% more GPU memory, and leads to a reduced training performance by ca. 20%. The TecoGAN$^{\ominus}$ model yields similar perceptual and temporal quality to DsDtPP with a significantly faster and more stable training.

Since the TecoGAN$^{\ominus}$ model requires less training resources, we also trained a larger generator with 50% more weights. In the following we will focus on this larger single-discriminator architecture with PP loss as our full *TecoGAN* model for VSR. Compared to the TecoGAN$^{\ominus}$ model, it can generate more details, and the training process is more stable, indicating that the larger generator and $D_{s,t}$ are more evenly balanced. Result images and temporal profiles are shown in Fig. 6 and Fig. 8. Video results are shown in Sec. 4 of the supplemental material.

We also carry out a similar ablation study for the UVT task. Again, we start from a single-image GAN-based model, a *CycleGAN* variant which already has two pairs of spatial generators and discriminators. Then, we train the *DsOnly* variant by adding flow estimation via $F$ and extending the spatial generators to frame-recurrent ones. By augmenting the two discriminators to use the triplet inputs proposed in Sec. 3, we arrive at the *Dst* model with spatio-temporal discriminators, which does not yet use the PP loss. Although UVT tasks are substantially different from VSR tasks, the comparisons in Fig. 1 and Sec. 4.6 of our supplemental material yield similar conclusions. In these tests, we use renderings of 3D fluid simulations of rising smoke as our unpaired training data. These simulations are generated with randomized numerical simulations using a resolution of $64^3$ for domain A and $256^3$ for domain B, and both are visualized with images of size $256^2$. Therefore, video translation from domain A to B is a tough task, as the latter contains significantly more turbulent and small-scale motions. With no temporal information available, the CycleGAN variant generates HR smoke that strongly flickers. The DsOnly model offers better temporal coherence by relying on its frame-recurrent input, but it learns a solution that largely ignores the current input and fails to keep reasonable spatio-temporal cycle-consistency links between the two domains. On the contrary, our $D_{s,t}$ enables the Dst model to learn the correlation between the spatial and temporal aspects, thus improving the cycle-consistency. However, without $\mathcal{L}_{pp}$, the Dst model (like the DsDt model of VSR) reinforces detail over time in an undesirable way. This manifests itself as inappropriate smoke density in empty regions. Using our full TecoGAN model which includes $\mathcal{L}_{pp}$, yields the best results, with detailed smoke structures and very good spatio-temporal cycle-consistency.

For comparison, a DsDtPP model involving a larger number of separate networks, i.e. four discriminators, two frame-recurrent generators and the $F$, is trained. By weighting the temporal adversarial losses from Dt with 0.3 and the spatial ones from Ds with 0.5, we arrived at a balanced training run. Although this model performs similarly to the TecoGAN model on the smoke dataset, the proposed spatio-temporal $D_{s,t}$ architecture represents a more preferable choice in practice, as it learns a natural balance of temporal and spatial components by itself, and requires fewer resources. Continuing along this direction, it will be interesting future work to evaluate variants, such as a shared $D_{s,t}$ for both domains, i.e. a multi-class classifier network.

Besides the smoke dataset, an ablation study for the Obama and Trump dataset from Fig. 1 shows a very similar behavior, as can be seen in the supplemental material.

**Spatio-temporal Adversarial Equilibriums** Our evaluation so far highlights that temporal adversarial learning is crucial for achieving spatial detail that is coherent over time for VSR, and for enabling the generators to learn the spatio-temporal correlation between domains in UVT. Next, we

Figure 7: Visual summary of VSR models. LPIPS (x-axis) measures spatial detail and temporal coherence is measured by tLP (y-axis) and tOF (bubble size with smaller as better). The middle graph zooms in the red-dashed-box region on the left, containing models in our ablation study. The right graph shows network sizes.

will shed light on the complex spatio-temporal adversarial learning objectives by varying the information provided to the discriminator network. The following tests $D_{s,t}$ networks that are identical apart from changing inputs, and we focus on the smoke dataset.

In order to learn the spatial and temporal features of the target domain as well as their correlation, the simplest input for $D_{s,t}$ consists of only the original, unwarped triplets, i.e. $\{I_g \text{ or } I_b\}$. Using these, we train a *baseline* model, which yields a sub-optimal quality: it lacks sharp spatial structures, and contains coherent but dull motions. Despite containing the full information, these input triplets prevent $D_{s,t}$ from providing the desired supervision. For paired video translation tasks, the *vid2vid* network achieves improved temporal coherence by using a video discriminator to supervise the output sequence conditioned with the ground-truth motion. With no ground-truth data available, we train a vid2vid variant by using the estimated motions and original triplets, i.e $\{I_g + F(g_{t-1}, g_t) + F(g_{t+1}, g_t) \text{ or } I_b + F(b_{t-1}, b_t) + F(b_{t+1}, b_t)\}$, as the input for $D_{s,t}$. However, the result do not significantly improve. The motions are only partially reliable, and hence don't help for the difficult unpaired translation task. Therefore, the discriminator still fails to fully correlate spatial and temporal features. We then train a third model, *concat*, using the original triplets and the warped ones, i.e. $\{I_g + I_{wg} \text{ or } I_b + I_{wb}\}$. In this case, the model learns to generate more spatial details with a more vivid motion. I.e., the improved temporal information from the warped triplets gives the discriminator important cues. However, the motion still does not fully resemble the target domain. We arrive at our final *TecoGAN* model for UVT by controlling the composition of the input data: as outlined above, we first provide only static triplets $\{I_{sg} \text{ or } I_{sb}\}$, and then apply the transitions of warped triplets $\{I_{wg} \text{ or } I_{wb}\}$, and original triplets $\{I_g \text{ or } I_b\}$ over the course of training. In this way, the network can first learn to extract spatial features, and build on them to establish temporal features. Finally, discriminators learn features about the correlation of spatial and temporal content by analyzing the original triplets, and provide gradients such that the generators learn to use the motion information from the input and  establish a correlation between the motions in the two unpaired domains. Consequently, the discriminator, despite receiving only a single triplet at once, can guide the generator to produce detailed structures that move coherently. Video comparisons are shown in Sec 5. of the supplemental material.

**Results and Metric Evaluation** While the visual results discussed above provide a first indicator of the quality our approach achieves, quantitative evaluations are crucial for automated evaluations across larger numbers of samples. Below we focus on the VSR task as ground-truth data is available in this case. We conduct user studies and present evaluations of the different models w.r.t. established spatial metrics. We also motivate and propose two novel temporal metrics to quantify temporal coherence. A visual summary is shown in Fig. 7.

For evaluating image SR, Blau & Michaeli (2018) demonstrated that there is an inherent trade-off between the perceptual quality of the result and the distortion measured with vector norms or low-level structures such as PSNR and SSIM. On the other hand, metrics based on deep feature maps such as LPIPS (Zhang et al., 2018) can capture more semantic similarities. We measure the PSNR and LPIPS using the Vid4 scenes. With a PSNR decrease of less than 2dB over DUF which has twice the model size of ours, TecoGAN outperforms all methods by more than 40% on LPIPS.

Table 2: Averaged VSR metric evaluations for the *Vid4* data set with the following metrics, PSNR: pixel-wise accuracy. LPIPS (AlexNet): perceptual distance to the ground truth. T-diff: pixel-wise differences of warped frames. tOF: pixel-wise distance of estimated motions. tLP: perceptual distance between consecutive frames. User study: Bradley-Terry scores (Bradley & Terry, 1952). Performance is averaged over 500 images up-scaled from 320x134 to 1280x536. More details can be found in Appendix B and C.

| Methods | PSNR↑ | LPIPS↓ ×10 | T-diff↓ ×100 | tOF↓ ×10 | tLP↓ ×100 | User Study ↑ | Model Size (M) ↓ | Processing Time (ms/frame) ↓ |
|---|---|---|---|---|---|---|---|---|
| DsOnly | 24.14 | 1.727 | 6.852 | 2.157 | 2.160 | - | 0.8(G)+1.7(F) | - |
| DsDt | 24.75 | 1.770 | 5.071 | 2.198 | 0.614 | - | 0.8(G)+1.7(F) | - |
| DsDtPP | 25.77 | 1.733 | 4.369 | 2.103 | **0.489** | - | 0.8(G)+1.7(F) | - |
| TecoGAN$^\ominus$ | 25.89 | 1.743 | 4.076 | 2.082 | 0.718 | - | 0.8(G)+1.7(F) | 37.07 |
| **TecoGAN** | 25.57 | **1.623** | 4.961 | 1.897 | 0.668 | **3.258** | 1.3(G)+1.7(F) | 41.92 |
| ENet | 22.31 | 2.458 | 9.281 | 4.009 | 4.848 | 1.616 | 0.8 | - |
| FRVSR | 26.91 | 2.506 | 3.648 | 2.090 | 0.957 | 2.600 | 0.8(SRNet)+1.7(F) | 36.95 |
| DUF | **27.38** | 2.607 | 3.298 | **1.588** | 1.329 | 2.933 | 6.2 | 942.21 |
| Bi-cubic | 23.66 | 5.036 | 3.152 | 5.578 | 2.144 | 0.0 | - | - |

Table 3: For the Obama&Trump dataset, the averaged tLP and tOF evaluations closely correspond to our user studies. The table below summarizes user preferences as Bradley-Terry scores.

| UVT scenes | Trump→Obama | | Obama→Trump | | AVG | | User Studies ↑, ref. to | |
|---|---|---|---|---|---|---|---|---|
| metrics | tLP↓ | tOF↓ | tLP↓ | tOF↓ | tLP↓ | tOF↓ | original input | arbitrary target |
| CycleGAN | 0.0176 | 0.7727 | 0.0277 | 1.1841 | 0.0234 | 0.9784 | 0.0 | 0.0 |
| RecycleGAN | **0.0111** | 0.8705 | 0.0248 | 1.1237 | 0.0179 | 0.9971 | 0.994 | 0.202 |
| TecoGAN | 0.0120 | **0.6155** | **0.0191** | **0.7670** | **0.0156** | **0.6913** | **1.817** | **0.822** |

While traditional temporal metrics based on vector norm differences of warped frames, e.g. T-diff, can be easily deceived by very blurry results, e.g. bi-cubic interpolated ones, we propose to use a tandem of two new metrics, tOF and tLP, to measure the consistence over time. tOF measures the pixel-wise difference of motions estimated from sequences, and tLP measures perceptual changes over time using deep feature map:

$$\text{tOF} = \|OF(b_{t-1}, b_t) - OF(g_{t-1}, g_t)\|_1 \quad \text{and} \quad \text{tLP} = \|LP(b_{t-1}, b_t) - LP(g_{t-1}, g_t)\|_1 \, , \quad (1)$$

where $OF$ represents an optical flow estimation with LucasKanade (1981) and $LP$ is the perceptual LPIPS metric. In tLP, the behavior of the reference is also considered, as natural videos exhibit a certain degree of changes over time. In conjunction, both pixel-wise differences and perceptual changes are crucial for quantifying realistic temporal coherence. While they could be combined into a single score, we list both measurements separately, as their relative importance could vary in different application settings. Our evaluation with these temporal metrics in Table 2 shows that all temporal adversarial models outperform spatial adversarial ones, and the full TecoGAN model performs very well: With a large amount of spatial detail, it still achieves good temporal coherence, on par with non-adversarial methods such as DUF and FRVSR. For VSR, we have confirmed these automated evaluations with several user studies. Across all of them, we find that the majority of the participants considered the TecoGAN results to be closest to the ground truth.

For the UVT tasks, where no ground-truth data is available, we can still evaluate tOF and tLP metrics by comparing the motion and the perceptual changes of the output data w.r.t. the ones from the input data , i.e., tOF $= \|OF(a_{t-1}, a_t) - OF(g_{t-1}^{a \to b}, g_t^{a \to b})\|_1$ and tLP$= \|LP(a_{t-1}, a_t) - LP(g_{t-1}^{a \to b}, g_t^{a \to b})\|_1$. With sharp spatial features and coherent motion, TecoGAN outperforms previous work on the Obama&Trump dataset, as shown in Table 3, although it is worth to point out that the tOF is less informative in this case, as the motion in the target domain is not necessarily pixel-wise aligned with the input. Overall, TecoGAN achieves good tLP scores thanks to its temporal coherence, on par with RecycleGAN, and its spatial detail is on par with CycleGAN. As for VSR, a perceptual evaluation by humans in the right column of Table 3 confirms our metric evaluations for the UVT task (details in Appendix C).

## 5 Conclusions and Discussion

In paired as well as unpaired data domains, we have demonstrated that it is possible to learn stable temporal functions with GANs thanks to the proposed discriminator architecture and PP loss. We have shown that this yields coherent and sharp details for VSR problems that go beyond what can be achieved with direct supervision. In UVT, we have shown that our architecture guides the training process to successfully establish the spatio-temporal cycle consistency between two domains. These results are reflected in the proposed metrics and user studies.

While our method generates very realistic results for a wide range of natural images, our method can generate temporally coherent yet sub-optimal details in certain cases such as under-resolved faces and text in VSR, or UVT tasks with strongly different motion between two domains. For the latter case, it would be interesting to apply both our method and motion translation from concurrent work (Chen et al., 2019). This can make it easier for the generator to learn from our temporal self supervision. In our method, the interplay of the different loss terms in the non-linear training procedure does not provide a guarantee that all goals are fully reached every time. However, we found our method to be stable over a large number of training runs, and we anticipate that it will provide a very useful basis for wide range of generative models for temporal data sets.

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

# APPENDIX

In the following, we first provide qualitative analysis(Appendix A) using multiple results that are mentioned but omitted in our main document due to space constraints. We then explain details of the metrics and present the quantitative analysis based on them(Appendix B). The conducted user studies are in support of our TecoGAN network and proposed temporal metrics (Appendix C). Then, we give technical details of our spatio-temporal discriminator (Sec. D), details of network architectures and training parameters (Appendix F, Appendix G). In the end, we discuss the performance of our approach in Appendix H.

## A QUALITATIVE ANALYSIS

For the VSR task, we test our model on a wide range of video data, including the generally used Vid4 dataset shown in Fig. 8 and 12, detailed scenes from the movie Tears of Steel (ToS, 2011) shown in Fig. 12, and others shown in Fig. 9. As mentioned in the main document, the TecoGAN model is trained with down-sampled inputs and it can similarly work with original images that were not down-sampled or filtered, such as a data-set of real-world photos (Liao et al., 2015). In Fig. 10, we compared our results to two other methods (Liao et al., 2015; Tao et al., 2017) that have used the same dataset. With the help of adversarial learning, our model is able to generate improved and realistic details in down-sampled images as well as captured images.

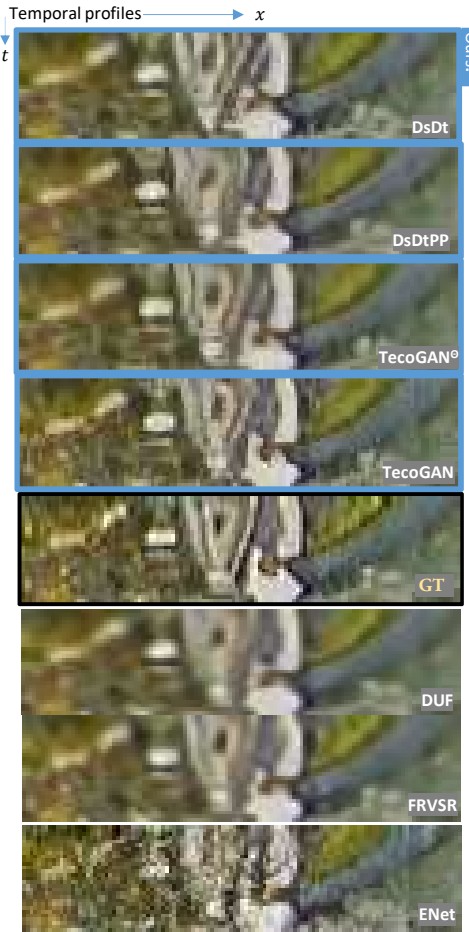

Figure 8: VSR temporal profile comparisons of the calendar scene (time shown along y-axis). Teco-GAN models lead to natural temporal progressions, and our final model closely matches the desired ground truth behavior over time.

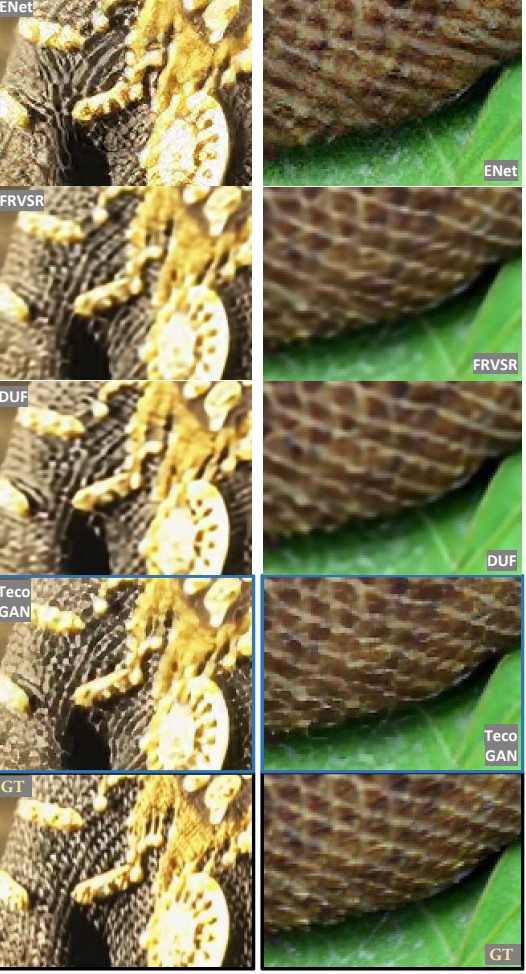

Figure 9: Additional VSR comparisons. The TecoGAN model generates sharp details in both scenes.

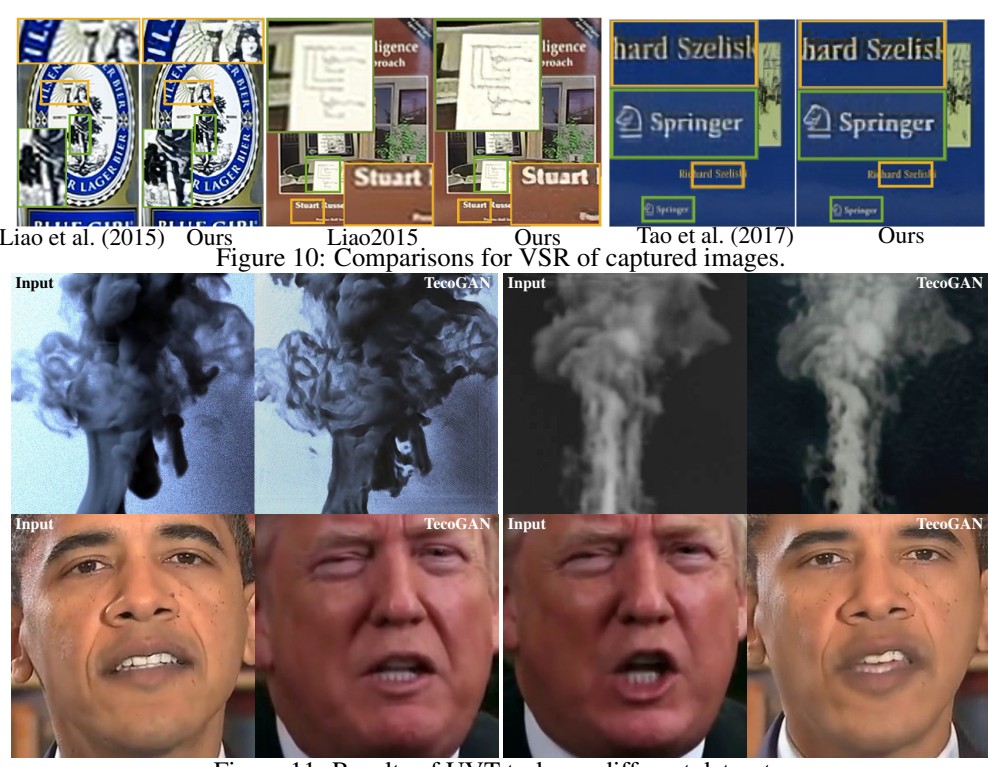

Liao et al. (2015)    Ours          Liao2015          Ours          Tao et al. (2017)    Ours
Figure 10: Comparisons for VSR of captured images.

Figure 11: Results of UVT tasks on different datasets.

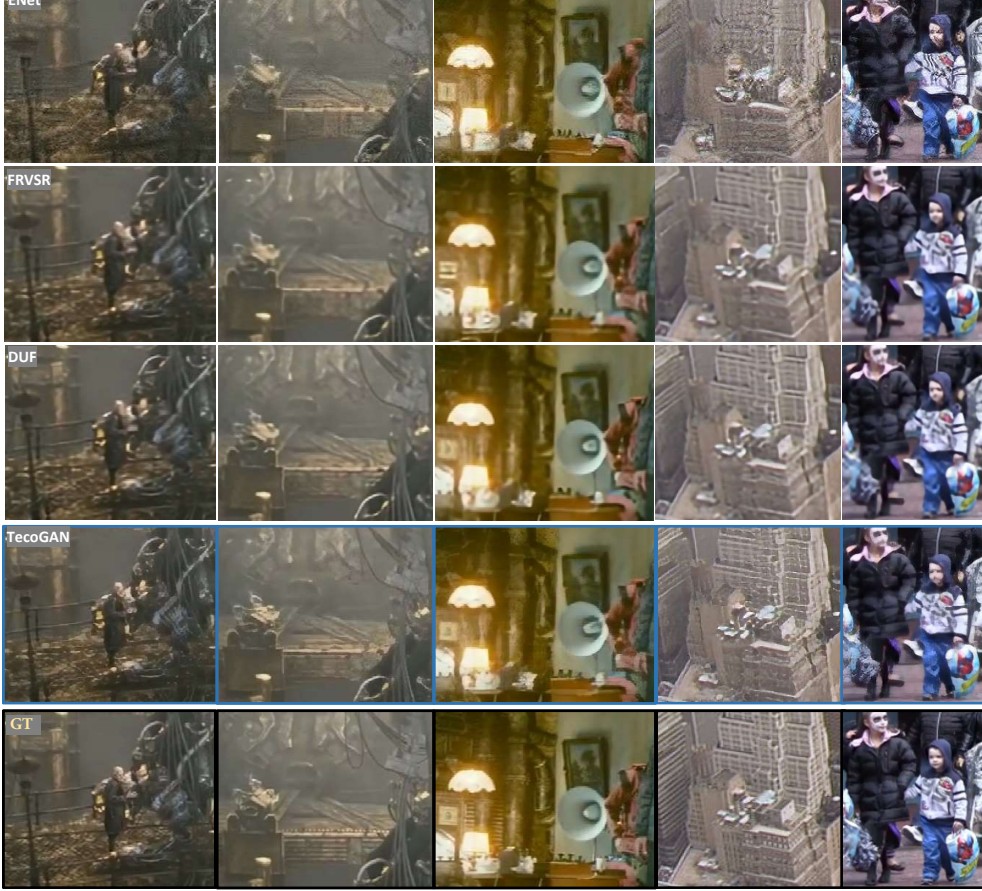

Figure 12: Detail views of the VSR results of ToS scenes (first three columns) and Vid4 scenes (two right-most columns) with comparisons.

Table 4: Metrics evaluated for the VSR Vid4 scenes.

| PSNR↑ | BIC | ENet | FRVSR | DUF | TecoGAN | TecoGAN⊖ | DsOnly | DsDt | DsDtPP | |
|---|---|---|---|---|---|---|---|---|---|---|
| calendar | 20.27 | 19.85 | 23.86 | 24.07 | 23.21 | 23.35 | 22.23 | 22.76 | 22.95 | |
| foliage | 23.57 | 21.15 | 26.35 | 26.45 | 24.26 | 25.13 | 22.33 | 22.73 | 25.00 | |
| city | 24.82 | 23.36 | 27.71 | 28.25 | 26.78 | 26.94 | 25.86 | 26.52 | 27.03 | |
| walk | 25.84 | 24.90 | 29.56 | 30.58 | 28.11 | 28.14 | 26.49 | 27.37 | 28.14 | |
| average | 23.66 | 22.31 | 26.91 | 27.38 | 25.57 | 25.89 | 24.14 | 24.75 | 25.77 | |
| LPIPS ↓×10 | BIC | ENet | FRVSR | DUF | TecoGAN | TecoGAN⊖ | DsOnly | DsDt | DsDtPP | |
| calendar | 5.935 | 2.191 | 2.989 | 3.086 | 1.511 | 2.142 | 1.532 | 2.111 | 2.112 | |
| foliage | 5.338 | 2.663 | 3.242 | 3.492 | 1.902 | 1.984 | 2.113 | 2.092 | 1.902 | |
| city | 5.451 | 3.431 | 2.429 | 2.447 | 2.084 | 1.940 | 2.120 | 1.889 | 1.989 | |
| walk | 3.655 | 1.794 | 1.374 | 1.380 | 1.106 | 1.011 | 1.215 | 1.057 | 1.051 | |
| average | 5.036 | 2.458 | 2.506 | 2.607 | 1.623 | 1.743 | 1.727 | 1.770 | 1.733 | |
| tOF ↓×10 | BIC | ENet | FRVSR | DUF | TecoGAN | TecoGAN⊖ | DsOnly | DsDt | DsDtPP | |
| calendar | 4.956 | 3.450 | 1.537 | 1.134 | 1.342 | 1.403 | 1.609 | 1.683 | 1.583 | |
| foliage | 4.922 | 3.775 | 1.489 | 1.356 | 1.238 | 1.444 | 1.543 | 1.562 | 1.373 | |
| city | 7.967 | 6.225 | 2.992 | 1.724 | 2.612 | 2.905 | 2.920 | 2.936 | 3.062 | |
| walk | 5.150 | 3.203 | 2.569 | 2.127 | 2.571 | 2.765 | 2.745 | 2.796 | 2.649 | |
| average | 5.578 | 4.009 | 2.090 | 1.588 | 1.897 | 2.082 | 2.157 | 2.198 | 2.103 | |
| tLP ↓×100 | BIC | ENet | FRVSR | DUF | TecoGAN | TecoGAN⊖ | DsOnly | DsDt | DsDtPP | |
| calendar | 3.258 | 2.957 | 1.067 | 1.603 | 0.165 | 1.087 | 0.872 | 0.764 | 0.670 | |
| foliage | 2.434 | 6.372 | 1.644 | 2.034 | 0.894 | 0.740 | 3.422 | 0.493 | 0.454 | |
| city | 2.193 | 7.953 | 0.752 | 1.399 | 0.974 | 0.347 | 2.660 | 0.490 | 0.140 | |
| walk | 0.851 | 2.729 | 0.286 | 0.307 | 0.653 | 0.635 | 1.596 | 0.697 | 0.613 | |
| average | 2.144 | 4.848 | 0.957 | 1.329 | 0.668 | 0.718 | 2.160 | 0.614 | 0.489 | |
| T-diff ↓×100 | BIC | ENet | FRVSR | DUF | TecoGAN | TecoGAN⊖ | DsOnly | DsDt | DsDtPP | GT |
| calendar | 2.271 | 9.153 | 3.212 | 2.750 | 4.663 | 3.496 | 6.287 | 4.347 | 4.167 | 6.478 |
| foliage | 3.745 | 11.997 | 3.478 | 3.115 | 5.674 | 4.179 | 8.961 | 6.068 | 4.548 | 4.396 |
| city | 1.974 | 7.788 | 2.452 | 2.244 | 3.528 | 2.965 | 4.929 | 3.525 | 2.991 | 4.282 |
| walk | 4.101 | 7.576 | 5.028 | 4.687 | 5.460 | 5.234 | 6.454 | 5.714 | 5.305 | 5.525 |
| average | 3.152 | 9.281 | 3.648 | 3.298 | 4.961 | 4.076 | 6.852 | 5.071 | 4.369 | 5.184 |

For UVT tasks, we train models for Obama and Trump translations, LR- and HR- smoke simulation translations, as well as translations between smoke simulations and real-smoke captures. While smoke simulations usually contain strong numerical viscosity with details limited by the simulation resolution, the real smoke, captured using the setup from Eckert et al. (2018), contains vivid fluid motions with many vortices and high-frequency details. As shown in Fig. 11, our method can be used to narrow the gap between simulations and real-world phenomenon.

## B  METRICS AND QUANTITATIVE ANALYSIS

**Spatial Metrics**   We evaluate all VSR methods with PSNR together with the human-calibrated LPIPS metric (Zhang et al., 2018). While higher PSNR values indicate a better pixel-wise accuracy, lower LPIPS values represent better perceptual quality and closer semantic similarity. Mean values of the Vid4 scenes Liu & Sun (2011) are shown on the top of Table 4. Trained with direct vector norms losses, FRVSR and DUF achieve high PSNR scores. However, the undesirable smoothing induced by these losses manifests themselves in larger LPIPS distances. ENet, on the other hand, with no information from neighboring frames, yields the lowest PSNR and achieves an LPIPS score that is only slightly better than DUF and FRVSR. TecoGAN model with adversarial training achieves an excellent LPIPS score, with a PSNR decrease of less than 2dB over DUF, which is very reasonable, since PSNR and perceptual quality were shown to be anti-correlated (Blau & Michaeli, 2018), especially in regions where PSNR is very high. Based on good perceptual quality and reasonable pixel-wise accuracy, TecoGAN outperforms all other methods by more than 40% for LPIPS.

**Temporal Metrics**   For both VSR and UVT, evaluating temporal coherence without ground-truth motion is a very challenging problem. The metric $T\text{-}diff = \|g_t - W(g_{t-1}, v_t)\|_1$ was used by Chen et al. (2017) as a rough assessment of temporal differences. As shown on bottom of Table 4, T-diff, due to its local nature, is easily deceived by blurry method such as the bi-cubic interrelation and

Table 5: Metrics evaluated for the VSR of ToS.

| PSNR↑ | BIC | ENet | FRVSR | DUF | TecoGAN | tOF ↓×10 | BIC | ENet | FRVSR | DUF | TecoGAN |
|---|---|---|---|---|---|---|---|---|---|---|---|
| room | 26.90 | 25.22 | 29.80 | 30.85 | 29.31 | room | 1.735 | 1.625 | 0.861 | 0.901 | 0.737 |
| bridge | 28.34 | 26.40 | 32.56 | 33.02 | 30.81 | bridge | 5.485 | 4.037 | 1.614 | 1.348 | 1.492 |
| face | 33.75 | 32.17 | 39.94 | 40.23 | 38.60 | face | 4.302 | 2.255 | 1.782 | 1.577 | 1.667 |
| average | 29.58 | 27.82 | 34.04 | 34.60 | 32.75 | average | 4.110 | 2.845 | 1.460 | 1.296 | 1.340 |
| LPIPS ↓×10 | BIC | ENet | FRVSR | DUF | TecoGAN | tLP ↓×100 | BIC | ENet | FRVSR | DUF | TecoGAN |
| room | 5.167 | 2.427 | 1.917 | 1.987 | 1.358 | room | 1.320 | 2.491 | 0.366 | 0.307 | 0.590 |
| bridge | 4.897 | 2.807 | 1.761 | 1.684 | 1.263 | bridge | 2.237 | 6.241 | 0.821 | 0.526 | 0.912 |
| face | 2.241 | 1.784 | 0.586 | 0.517 | 0.590 | face | 1.270 | 1.613 | 0.290 | 0.314 | 0.379 |
| average | 4.169 | 2.395 | 1.449 | 1.414 | 1.086 | average | 1.696 | 3.827 | 0.537 | 0.403 | 0.664 |

can not correlate well with visual assessments of coherence. By measuring the pixel-wise motion difference using tOF in together with the perceptual changes over time using tLP, we show the temporal evaluations for the VSR task in the middle of Table 4. Not surprisingly, the results of ENet show larger errors for all metrics due to their strongly flickering content. Bi-cubic up-sampling, DUF, and FRVSR achieve very low T-diff errors due to their smooth results, representing an easy, but undesirable avenue for achieving coherency. However, the overly smooth changes of the former two are identified by the tLP scores.While our DsOnly model generates sharper results at the expense of temporal coherence, it still outperforms ENet there. By adding temporal information to discriminators, our DsDt, DsDt+PP, TecoGAN$^\ominus$and TecoGAN improve in terms of temporal metrics. Especially the full TecoGAN model stands out here. For the UVT tasks, temporal motions are evaluated by comparing to the input sequence. With sharp spatial features and coherent motion, TecoGAN outperforms previous work on the Obama&Trump dataset, as shown in Table 3.

**Spatio-temporal Evaluations** Since temporal metrics can trivially be reduced for blurry image content, we found it important to evaluate results with a combination of spatial and temporal metrics. Given that perceptual metrics are already widely used for image evaluations, we believe it is the right time to consider perceptual changes in temporal evaluations, as we did with our proposed temporal coherence metrics. Although not perfect, they are not easily deceived. Specifically, tOF is more robust than a direct pixel-wise metric as it compares motions instead of image content. In the supplemental material, we visualize the motion difference and it can well reflect the visual inconsistencies. On the other hand, we found that our calculation of tLP is a general concept that can work reliably with different perceptual metric: When repeating the tLP evaluation with the PieAPP metric (Prashnani et al., 2018) instead of $LP$, i.e., $tPieP = \|f(y_{t-1}, y_t) - f(g_{t-1}, g_t)\|_1$ , where f(·) indicates the perceptual error function of PieAPP, we get close to identical results, listed in Fig. 13. The conclusions from $tPieP$ also closely match the LPIPS-based evaluation: our network architecture can generate realistic and temporally coher-

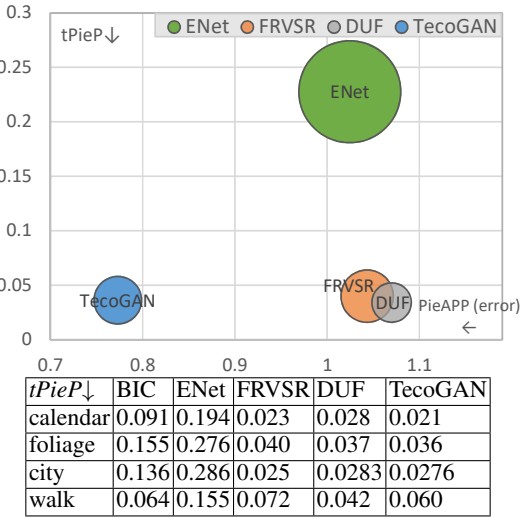

| $tPieP$↓ | BIC | ENet | FRVSR | DUF | TecoGAN |
|---|---|---|---|---|---|
| calendar | 0.091 | 0.194 | 0.023 | 0.028 | 0.021 |
| foliage | 0.155 | 0.276 | 0.040 | 0.037 | 0.036 |
| city | 0.136 | 0.286 | 0.025 | 0.0283 | 0.0276 |
| walk | 0.064 | 0.155 | 0.072 | 0.042 | 0.060 |

Figure 13: Tables and visualization of perceptual metrics computed with PieAPP (Prashnani et al., 2018) (instead of LPIPS used in Fig. 7 previously) on ENet, FRVSR, DUF and TecoGAN for the VSR of Vid4. Bubble size indicates the tOF score.

ent detail, and the metrics we propose allow for a stable, automated evaluation of the temporal perception of a generated video sequence.

Besides the previously evaluated the Vid4 dataset, with graphs shown in Fig. 14, 15, we also get similar evaluation results on the *Tears of Steel* data-sets (room, bridge, and face, in the following referred to as *ToS* scenes) and corresponding results are shown in Table 5 and Fig. 16. In all tests, we follow the procedures of previous work (Jo et al., 2018; Sajjadi et al., 2018) to make the outputs of

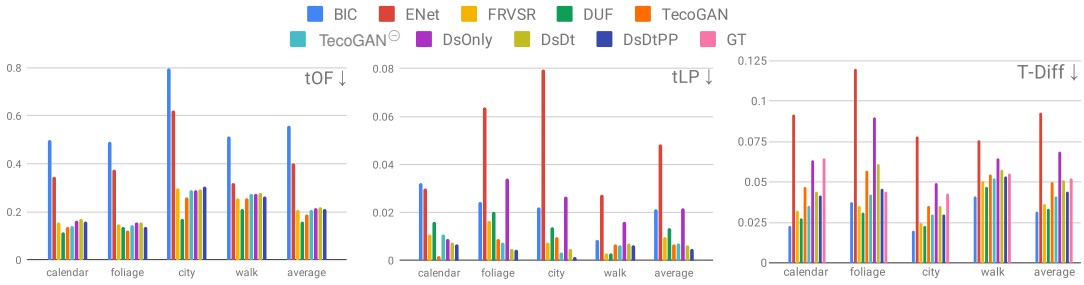

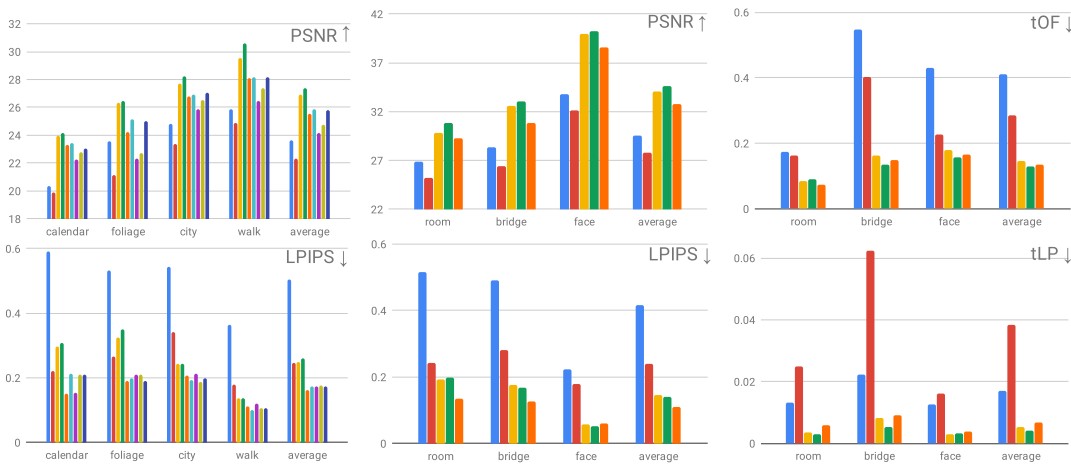

Figure 14: Bar graphs of temporal metrics for Vid4.

Figure 15: Spatial metrics for Vid4.

Figure 16: Metrics for ToS.

all methods comparable, i.e., for all result images, we first exclude spatial borders with a distance of 8 pixels to the image sides, then further shrink borders such that the LR input image is divisible by 8 and for spatial metrics, we ignore the first two and the last two frames, while for temporal metrics, we ignore first three and last two frames, as an additional previous frame is required for inference. In the following, we conduct user studies for the Vid4 scenes. By comparing the user study results and the metric breakdowns shown in Table 4, we found our metrics to reliably capture the human temporal perception, as shown in Appendix C.

## C   USER STUDIES

We conduct several user studies for the VSR task using five different methods, namely bi-cubic interpolation, ENet, FRVSR, DUF and our TecoGAN. The established 2AFC design (Fechner & Wundt, 1889; Um et al., 2017) is applied, i.e., participants have a pair-wise choice, with the ground-truth video shown as reference. One example can be seen in Fig. 17. The videos are synchronized and looped until user made the final decision. With no control to stop videos, users Participants cannot stop or influence the playback, and hence can focus more on the whole video, instead of specific spatial details. Videos positions (left/A or right/B) are randomized.

After collecting 1000 votes from 50 users for every scene, i.e. twice for all possible pairs ($5 \times 4/2 = 10$ pairs), we follow common procedure and compute scores for all models with the Bradley-Terry model (1952). The outcomes for the Vid4 scenes can be seen in Fig. 18 (overall scores are listed in Table 2 of the main document).

From the Bradley-Terry scores for the Vid4 scenes we can see that the TecoGAN model performs very well, and achieves the first place in three cases, as well as a second place in the walk scene. The latter is most likely caused by the overall slightly smoother images of the walk scene, in conjunction with the presence of several human faces, where our model can lead to the generation of unexpected

details. However, overall the user study shows that users preferred the TecoGAN output over the other two deep-learning methods with a 63.5% probability.

This result also matches with our metric evaluations. In Table 4, while TecoGAN achieves spatial (LPIPS) improvements in all scenes, DUF and FRVSR are not far behind in the walk scene. In terms of temporal metrics tOF and tLP, TecoGAN achieves similar or lower scores compared to FRVSR and DUF for calendar, foliage and city scenes. The lower performance of our model for the walk scene is likewise captured by higher tOF and tLP scores. Overall, the metrics confirm the performance of our TecoGAN approach and match the results of the user studies, which indicate that our proposed temporal metrics successfully capture important temporal aspects of human perception.

For UVT tasks which have no ground-truth data, we carried out two sets of user studies: One uses an arbitrary sample from the target domain as the reference and the other uses the actual input from the source domain as the reference. On the Obama&Trump data-sets, we evaluate results from CycleGAN, RecycleGAN, and TecoGAN following the same modality, i.e. a 2AFC design with 50 users for each run. E.g., on the left of Fig. 19, users evaluate the generated Obama in reference with the input Trump on the y-axis, while an arbitrary Obama video is shown as the reference on the x-axis. Effectively, the y-axis is more important than the x-axis as it indicates whether the translated result preserves the original expression. A consistent ranking of TecoGAN > RecycleGAN > CycleGAN is shown on the y-axis with clear separations, i.e. standard errors don't overlap. The x-axis indicates whether the inferred result matches the general spatio-temporal content of the target domain. Our TecoGAN model also receives the highest scores here, although the responses are slightly more spread out. On the right of Fig. 19, we summarize both studies in a single graph highlighting that the TecoGAN model is consistently preferred by the participants of our user studies.

## D    TECHNICAL DETAILS OF THE SPATIO-TEMPORAL DISCRIMINATOR

**Motion Compensation Used in Warped Triplet**  In the TecoGAN architecture, $D_{s,t}$ detects the temporal relationships between $IN_{s,t}^g$ and $IN_{s,t}^y$ with the help of the flow estimation network F. However, at the boundary of images, the output of F is usually less accurate due to the lack of reliable neighborhood information. There is a higher chance that objects move into the field of view, or leave suddenly, which significantly affects the images warped with the inferred motion. An example is shown in Fig. 20. This increases the difficulty for $D_{s,t}$, as it cannot fully rely on the images being aligned via warping.    To alleviate this problem, we only use the center region of $IN_{s,t}^g$ and $IN_{s,t}^y$ as the input of the discriminator, and we reset a boundary of 16 pixels. Thus, for an input resolution of $IN_{s,t}^g$ and $IN_{s,t}^y$ of

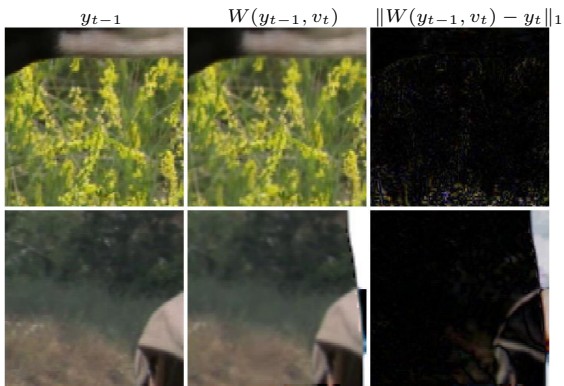

Figure 20: Near image boundaries, flow estimation is less accurate and warping often fails to align well. First two columns show original and warped frames and the third one shows differences after warping (ideally all black). The top row shows things move into the view with problems near lower boundaries, while the second row has objects moving out of the view.

$128 \times 128$ for the VSR task, the inner part in size of $96 \times 96$ is left untouched, while the border regions are overwritten with zeros.

The flow estimation network F with the loss $\mathcal{L}_{G,F}$ should only be trained to support G in reaching the output quality as determined by $D_{s,t}$, but not the other way around. The latter could lead to F networks that confuse $D_{s,t}$ with strong distortions of $IN_{s,t}^g$ and $IN_{s,t}^y$. In order to avoid the this undesirable case, we stop the gradient back propagation from $IN_{s,t}^g$ and $IN_{s,t}^y$ to F. In this way, gradients from $D_{s,t}$ to F are only back propagated through the generated samples $g_{t-1}$, $g_t$ and $g_{t+1}$ into the generator network. In this way $D_{s,t}$ can guide G to improve the image content, and F learns

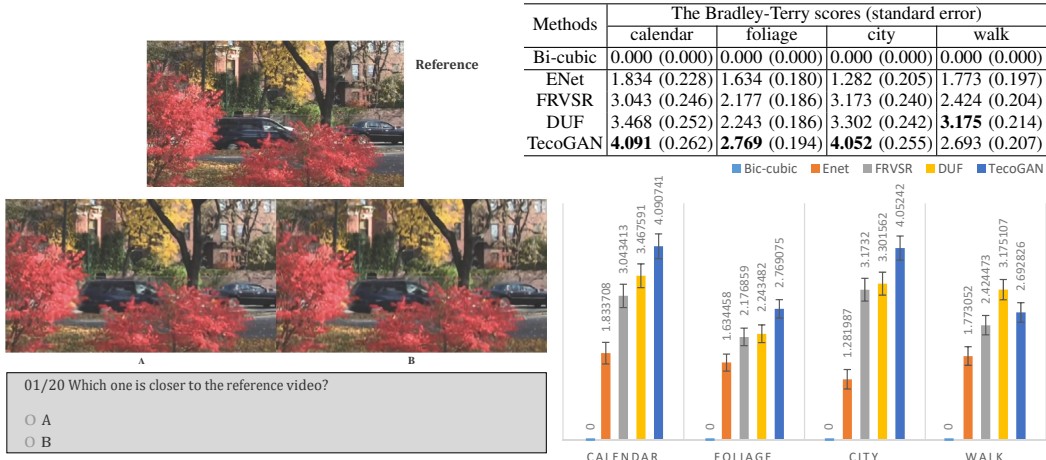

Figure 17: A sample setup of user study.

Figure 18: Tables and bar graphs of Bradley-Terry scores and standard errors for Vid4 VSR.

| The Bradley-Terry scores (standard error) | | | | |
|---|---|---|---|---|
| Methods | calendar | foliage | city | walk |
| Bi-cubic | 0.000 (0.000) | 0.000 (0.000) | 0.000 (0.000) | 0.000 (0.000) |
| ENet | 1.834 (0.228) | 1.634 (0.180) | 1.282 (0.205) | 1.773 (0.197) |
| FRVSR | 3.043 (0.246) | 2.177 (0.186) | 3.173 (0.240) | 2.424 (0.204) |
| DUF | 3.468 (0.252) | 2.243 (0.186) | 3.302 (0.242) | **3.175** (0.214) |
| TecoGAN | **4.091** (0.262) | **2.769** (0.194) | **4.052** (0.255) | 2.693 (0.207) |

| X-axis: Generated Obama vs. Obama | Bradley-Terry scores | Std. Error | X-axis: Generated Obama vs. Obama | Bradley-Terry scores | Std. Error | X-axis: Generated Obama vs. Obama | Bradley-Terry scores | Std. Error |
|---|---|---|---|---|---|---|---|---|
| CycleGAN | 0 | 0 | CycleGAN | 0.806 | 0.177 | CycleGAN | 0 | 0 |
| RecycleGAN | 1.322 | 0.197 | RecycleGAN | 0 | 0 | RecycleGAN | 0.202 | 0.118 |
| TecoGAN | 1.520 | 0.201 | TecoGAN | 1.727 | 0.208 | TecoGAN | 0.822 | 0.123 |
| Y-axis: Generated Obama vs. Trump | Bradley-Terry scores | Std. Error | X-axis: Generated Obama vs. Obama | Bradley-Terry scores | Std. Error | X-axis: Generated Obama vs. Obama | Bradley-Terry scores | Std. Error |
| CycleGAN | 0 | 0 | CycleGAN | 0 | 0 | CycleGAN | 0 | 0 |
| RecycleGAN | 1.410 | 0.208 | RecycleGAN | 0.623 | 0.182 | RecycleGAN | 0.994 | 0.135 |
| TecoGAN | 1.958 | 0.222 | TecoGAN | 1.092 | 0.182 | TecoGAN | 1.817 | 0.150 |

Figure 19: Tables and graphs of Bradley-Terry scores and standard errors for Obama&Trump UVT.

to warp the previous frame in accordance with the detail that G can synthesize. However, F does not adjust the motion estimation only to reduce the adversarial loss.

**Curriculum Learning for UVT Discriminators**   As mentioned in the main part, we train the UVT $D_{s,t}$ with 100% spatial triplets at the very beginning. During training, 25% of them gradually transfer into warped triplets and another 25% transfer into original triplets. The transfer of the warped triplets can be represented as: $(1-\alpha)\mathrm{I}_{cg}+\alpha\mathrm{I}_{wg}$, with $\alpha$ growing form 0 to 1. For the original triplets, we additionally fade the "warping" operation out by using $(1 - \alpha)\mathrm{I}_{cg} + \alpha\{W(g_{t-1}, v_t * \beta), g_t, W(g_{t+1}, v'_t * \beta)\}$, again with $\alpha$ growing form 0 to 1 and $\beta$ decreasing from 1 to 0. We found this smooth transition to be helpful for a stable training.

# E   DATA AUGMENTATION AND TEMPORAL CONSTRAINS IN THE PP LOSS

Since training with sequences of arbitrary length is not possible with current hardware, problems such as the streaking artifacts discussed above generally arise for recurrent models. In the proposed PP loss, both the Ping-Pang data augmentation and the temporal consistency constraint contribute to solving these problems. In order to show their separated contributions, we trained another TecoGAN variant that only employs the data augmentation without the constraint (i.e., $\lambda_p = 0$ in Table 1).

Denoted as PP-Augment, we show its results in comparison with the DsDt and TecoGAN$^\ominus$ models in Fig. 21. Video results are shown in the in the supplemental material.

During training, the generator of DsDt receives 10 frames, and generators of PP-Augment and TecoGAN$^\ominus$ see 19 frames. While DsDt shows strong recurrent accumulation artifacts early on, the PP-Augment version slightly reduces the artifacts. In Fig. 21, it works good for frame 15, but shows artifacts from frame 32 on. Only our regular model (TecoGAN$^\ominus$) successfully avoids temporal accumulation for all 40 frames. Hence, with the PP constraint, the model avoids recurrent accumulation of artifacts and works well for sequences that are substantially longer than the training length.

| DsDt | PP-Augment | TecoGAN$^\ominus$ |
|---|---|---|
| PSNR: 24.75, LPIPS: 1.77, tLP: 0.614, tOF 2.198 | PSNR: 24.98, LPIPS: 1.81, tLP: 0.850, tOF 1.903 | PSNR: 25.89, LPIPS: 1.74, tLP: 0.718, tOF 2.082 |

Figure 21: 1st & 2nd row: Frame 15 & 40 of the *Foliage* scene. While DsDt leads to strong recurrent artifacts early on, PP-Augment shows similar artifacts later in time (2nd row, middle). TecoGAN$^\ominus$ model successfully removes these artifacts.

Among others, we have tested our model with ToS sequences of lengths 150, 166 and 233. For all of these sequences, the TecoGAN model successfully avoids temporal accumulation or streaking artifacts.

## F  NETWORK ARCHITECTURE

In this section, we use the following notation to specify all network architectures used: conc() represents the concatenation of two tensors along the channel dimension; $C/CT$(input, kernel_size, output_channel, stride_size) stands for the convolution and transposed convolution operation, respectively; "+" denotes element-wise addition; BilinearUp2 up-samples input tensors by a factor of 2 using bi-linear interpolation; BicubicResize4(input) increases the resolution of the input tensor to 4 times higher via bi-cubic up-sampling; $Dense$(input, output_size) is a densely-connected layer, which uses Xavier initialization for the kernel weights.

The architecture of our VSR generator G is:

$$\text{conc}(x_t, W(g_{t-1}, v_t)) \to l_{in} \,; C(l_{in}, 3, 64, 1), \text{ReLU} \to l_0;$$
$$ResidualBlock(l_i) \to l_{i+1} \text{ with } i = 0, ..., n-1;$$
$$CT(l_n, 3, 64, 2), \text{ReLU} \to l_{up2}; CT(l_{up2}, 3, 64, 2), \text{ReLU} \to l_{up4};$$
$$C(l_{up4}, 3, 3, 1), \text{ReLU} \to l_{res}; \text{BicubicResize4}(x_t) + l_{res} \to g_t \,.$$

In TecoGAN$^\ominus$, there are 10 sequential residual blocks in the generator ( $l_n = l_{10}$ ), while the TecoGAN generator has 16 residual blocks ( $l_n = l_{16}$ ). Each $ResidualBlock(l_i)$ contains the following operations: $C(l_i, 3, 64, 1), \text{ReLU} \to r_i; C(r_i, 3, 64, 1) + l_i \to l_{i+1}$.

The VSR $D_{s,t}$'s architecture is:

$$\text{IN}_{s,t}^g \text{ or IN}_{s,t}^y \to l_{in}; C(l_{in}, 3, 64, 1), \text{Leaky ReLU} \to l_0;$$
$$C(l_0, 4, 64, 2), \text{BatchNorm}, \text{Leaky ReLU} \to l_1; C(l_1, 4, 64, 2), \text{BatchNorm}, \text{Leaky ReLU} \to l_2;$$
$$C(l_2, 4, 128, 2), \text{BatchNorm}, \text{Leaky ReLU} \to l_3; C(l_3, 4, 256, 2), \text{BatchNorm}, \text{Leaky ReLU} \to l_4;$$
$$Dense(l_4, 1), \text{sigmoid} \to l_{out} \,.$$

VSR discriminators used in our variant models, DsDt, DsDtPP and DsOnly, have a similar architecture as $D_{s,t}$. They only differ in terms of their inputs.

The flow estimation network F has the following architecture:

$$\text{conc}(x_t, x_{t-1}) \to l_{in}; C(l_{in}, 3, 32, 1), \text{Leaky ReLU} \to l_0;$$
$$C(l_0, 3, 32, 1), \text{Leaky ReLU}, \text{MaxPooling} \to l_1; C(l_1, 3, 64, 1), \text{Leaky ReLU} \to l_2;$$
$$C(l_2, 3, 64, 1), \text{Leaky ReLU}, \text{MaxPooling} \to l_3; C(l_3, 3, 128, 1), \text{Leaky ReLU} \to l_4;$$
$$C(l_4, 3, 128, 1), \text{Leaky ReLU}, \text{MaxPooling} \to l_5; C(l_5, 3, 256, 1), \text{Leaky ReLU} \to l_6;$$
$$C(l_6, 3, 256, 1), \text{Leaky ReLU}, \text{BilinearUp2} \to l_7; C(l_7, 3, 128, 1), \text{Leaky ReLU} \to l_8;$$

$$C(l_8, 3, 128, 1), \text{Leaky ReLU}, \text{BilinearUp2} \rightarrow l_9; C(l_9, 3, 64, 1), \text{Leaky ReLU} \rightarrow l_{10};$$
$$C(l_{10}, 3, 64, 1), \text{Leaky ReLU}, \text{BilinearUp2} \rightarrow l_{11}; C(l_{11}, 3, 32, 1), \text{Leaky ReLU} \rightarrow l_{12};$$
$$C(l_{12}, 3, 2, 1), \tanh \rightarrow l_{out}; l_{out} * \text{MaxVel} \rightarrow v_t \ .$$

Here, MaxVel is a constant vector, which scales the network output to the normal velocity range.

While F is the same for UVT tasks, UVT generators have an encoder-decoder structure:

$$\text{conc}(x_t, W(g_{t-1}, v_t)) \rightarrow l_{in}; C(l_{in}, 7, 32, 1), \text{InstanceNorm}, \text{ReLU} \rightarrow l_0;$$
$$C(l_0, 3, 64, 2), \text{InstanceNorm}, \text{ReLU} \rightarrow l_1; C(l_1, 3, 128, 2), \text{InstanceNorm}, \text{ReLU} \rightarrow l_2;$$
$$ResidualBlock(l_2 + i) \rightarrow l_{3+i} \text{ with } i = 0, ..., n - 1;$$
$$CT(l_{n+2}, 3, 64, 2), \text{InstanceNorm}, \text{ReLU} \rightarrow l_{n+3}; CT(l_{n+3}, 3, 32, 2), \text{InstanceNorm}, \text{ReLU} \rightarrow l_{n+4};$$
$$CT(l_{n+4}, 7, 3, 1), \tanh \rightarrow l_{out}$$

$ResidualBlock(l_2 + i)$ contains the following operations: $C(l_{2+i}, 3, 128, 1), \text{InstanceNorm}, \text{ReLU} \rightarrow t_{2+i}$ ;$C(t_{2+i}, 3, 128, 1), \text{InstanceNorm} \rightarrow r_{2+i}; r_{2+i} + l_{2+i} \rightarrow l_{3+i}$. We use 10 residual blocks for all UVT generators.

Since UVT generators are larger than the VSR generator, we also use a larger $D_{s,t}$ architecture:

$$\text{IN}_{s,t}^g \text{ or } \text{IN}_{s,t}^y \rightarrow l_{in}; C(l_{in}, 4, 64, 24), \text{ReLU} \rightarrow l_0;$$
$$C(l_0, 4, 128, 2), \text{InstanceNorm}, \text{Leaky ReLU} \rightarrow l_1; C(l_1, 4, 256, 2), \text{InstanceNorm}, \text{Leaky ReLU} \rightarrow l_2;$$
$$C(l_2, 4, 512, 2), \text{InstanceNorm}, \text{Leaky ReLU} \rightarrow l_3; Dense(l_3, 1) \rightarrow l_{out} \ .$$

Again, all ablation studies use the same architecture with different inputs.

## G  TRAINING DETAILS

We use the non-saturated GAN for VSR and LSGAN (Mao et al., 2017) for UVT and both of them can prevent the gradient vanishing problem of a vanilla GAN (Goodfellow et al., 2014). While we train stably with a dynamic discriminator updating strategy, i.e. discriminators are not updated when there is already a large difference between $D(\text{I}^b)$ and $D(\text{I}^g)$, the training process could potentially be further improved with modern GAN algorithms, e.g. Wasserstein GAN (Gulrajani et al., 2017).We train G and $F$ together for VSR , while we simply use the pre-trained $F$ for UVT.

For the VSR task, our training data-set consists of 250 short HR videos, each with 120 frames. We use sequences with a length of 10 and a batch size of 4. A black image is used as the first previous frame of each video sequence. I.e., one batch contains 40 frames and with the PP loss formulation, the NN receives gradients from 76 frames in total for every training iteration. To improve the stability of the adversarial training, we pre-train G and $F$ with a simple $L^2$ loss of $\sum \|g_t - b_t\|_2 + \lambda_w \mathcal{L}_{warp}$ for 500k batches. We use 900k batches for the adversarial training stage. The data-sets of the UVT tasks contain around 2400 to 3600 frames. We train the generators with a sequence length of 6 and a batch size of 1. Since temporal triplets are gradually faded in, we do not pre-train models for UVT tasks. With smaller datasets, we train UVT models with 100k batches.

In the pre-training stage of VSR, we train the F and a generator with 10 residual blocks. An ADAM optimizer with $\beta = 0.9$ is used throughout. The learning rate starts from $10^{-4}$ and decays by 50% every 50k batches until it reaches $2.5 * 10^{-5}$. This pre-trained model is then used for all TecoGAN variants as initial state. In the adversarial training stage of VSR, all TecoGAN variants are trained with a fixed learning rate of $5 * 10^{-5}$. The generators in DsOnly, DsDt, DsDtPP and TecoGAN$^\ominus$ have 10 residual blocks, whereas the TecoGAN model has 6 additional residual blocks in its generator. Therefore, after loading 10 residual blocks from the pre-trained model, these additional residual blocks are faded in smoothly with a factor of $2.5 * 10^{-5}$. We found this growing training methodology, first introduced by Growing GAN (Karras et al., 2017), to be stable and efficient in our tests. When training the VSR DsDt and DsDtPP, extra parameters are used to balance the two cooperating discriminators properly. Through experiments, we found $D_t$ to be stronger. Therefore, we reduce the learning rate of $D_t$ to $1.5 * 10^{-5}$ in order to keep both discriminators balanced. At the same time, a factor of 0.0003 is used on the temporal adversarial loss to the generator, while the spatial adversarial loss has a factor of 0.001. During the VSR training, input LR video frames are cropped to a size of $32 \times 32$. In all VSR models, the Leaky ReLU operation uses a tangent of 0.2 for the negative half space. Additional training parameters are listed in Table 6.

Table 6: Training parameters

| VSR Param | DsOnly | DsDt | DsDtPP | TecoGAN⊖ | TecoGAN |
|---|---|---|---|---|---|
| $\lambda_a$ | 1e-3 | Ds: 1e-3, Dt: 3e-4 | 1e-3 | | 1e-3 |
| $\lambda_p$ | 0.0 | 0.0 | 0.5 | | |
| $\lambda_\phi$ | 0.02 for VGG and 1.0 for Discriminator | | | | |
| $\lambda_\omega, \lambda_c$ | 1.0, 1.0 | | | | |
| learning-rate | 5e-5 | 1.5e-5 for Dt, 5e-5 for others. | 5e-5 | | 5e-5 |

| UVT Param | DsOnly | Dst | DsDtPP | TecoGAN |
|---|---|---|---|---|
| $\lambda_a$ | 0.5 | | Ds: 0.5 Dt: 0.3 | 0.5 |
| $\lambda_p$ | 0.0 | 0.0 | 100.0 | |
| $\lambda_\phi$ | from $10^6$ decays to 0.0 | | | |
| $\lambda_\omega$ | 0.0, a pre-trained F is used for UST tasks | | | |
| $\lambda_c$ | 10.0 | | | |

For all UVT tasks, we use a learning rate of $10^{-4}$ to train the first 90k batches and the last 10k batches are trained with the learning rate decay from $10^{-4}$ to 0. Images of the input domain are cropped into a size of $256 \times 256$ when training, while the original size is $288 \times 288$. While the Additional training parameters are also listed in Table 6. For UVT, $\mathcal{L}_{\text{content}}$ and $\mathcal{L}_\phi$ are only used to improve the convergence of the training process. We fade out the $\mathcal{L}_{\text{content}}$ in the first 10k batches and the $\mathcal{L}_\phi$ is used for the first 80k and faded out in last 20k.

## H  PERFORMANCE

TecoGAN is implemented in TensorFlow. While generator and discriminator are trained together, we only need the trained generator network for the inference of new outputs after training, i.e., the whole discriminator network can be discarded. We evaluate the models on a Nvidia GeForce GTX 1080Ti GPU with 11G memory, the resulting VSR performance for which is given in Table 2.

The VSR TecoGAN⊖ model and FRVSR have the same number of weights (843587 in the SRNet, i.e. generator network, and 1.7M in F), and thus show very similar performance characteristics with around 37 ms spent for one frame. The larger VSR TecoGAN model with 1286723 weights in the generator is slightly slower than TecoGAN⊖, spending 42 ms per frame. In the UVT task, generators spend around 60 ms per frame with a size of $512 \times 512$. However, compared with the DUF model, with has more than 6 million weights in total, the TecoGAN performance significantly better thanks to its reduced size.

