# OpenReview forum: "Learning Temporal Coherence via Self-Supervision for GAN-based Video Generation"
_ICLR.cc/2020/Conference — Reject_

### Official Review · AnonReviewer1 · 2019-10-23
**Official Blind Review #1**

**Rating:** 3

**Review:**

Contribution
Augments the loss of video generation systems with a discriminator that considers multiple frames (as opposed to single frames independently) and a new objective termed ping-pong loss which is introduced in order to deal with “artifacts” that appear in video generation. The paper also proposes a few automatic metrics with which to compare systems. Although the performance does not convincingly exceed its competitors, the contribution seems to be getting the spatio-temporal adversarial loss to work at all.

Overall
I found the video generations impressive, and the addition of the discriminator seems to improve sharpness of the individual frames over methods trained with non-adversarial losses, in particular DUF. Although TecoGAN does not beat its competitors (for example DUF) on most proposed automatic metrics, it appears to be ranked better in expectation in user studies (Table 2). However, I am concerned about generalizability of the method.

Decision
As someone not in this field, I find it hard to judge the performance of the system from so few samples. I also found the differences in the losses between tasks to be worrisome, as it indicates this method does not generalize without heavy tweaking of the loss. My current decision is weak reject, since the generations look quite good but I have concerns about generalization of the approach to other video generation tasks as well as the justification of the ping-pong loss.

Questions
- Is RecycleGAN unable to be applied to video super-resolution? Why is it not compared to in Table 2?
- Is L_Phi the perceptual loss? I don't think it was mentioned in the body of the text.
- I found the ping-pong (PP) loss to be unjustified. The loss is motivated by the issue of "artifacts", which I assume are poor generations due to lacking a good model of the world. The paper says this issue could be alleviated by training with longer video sequences, but that would prevent the generator from working with sequences of arbitrary length. I do not believe the ping-pong loss allows the generator to work with arbitrary sequences, as training only on short sequences and their reverse should not allow the model to generalize to longer sequences. Additionally, this approach would likely not scale to long sequences as well, since it requires doubling the sequence length. Would you be able to train on longer sequences?

**Experience Assessment:**

I do not know much about this area.

**Review Assessment: Checking Correctness Of Derivations And Theory:**

N/A

**Review Assessment: Checking Correctness Of Experiments:**

I assessed the sensibility of the experiments.

**Review Assessment: Thoroughness In Paper Reading:**

I read the paper at least twice and used my best judgement in assessing the paper.

---

> ### Author Response · Authors · 2019-11-12
> **Reply to reviewer #1 (part 1)**
>
> Dear reviewer, thank you very much for the detailed review. Among others, we were glad to hear that the video results were considered to be impressive.
>
> Q1: “…the contribution seems to be getting the spatio-temporal adversarial loss to work at all ...”
>
> A1: For image generation, GANs are extremely popular in order to synthesize realistic results. For videos, which add an additional dimension to the data domain, GANs are scarcely used due to a lack of techniques to control temporal coherence. We believe that showing how spatio-temporal adversarial learning can yield realistic and coherent results is an important step in this field. Many highly successful papers, e.g., the growing GANs of Karras et al., have introduced simple concepts that have led to huge advances.
>
> Due to the inherent challenges of GAN-training we also found it important to demonstrate how spatio-temporal adversarial learning can be realized in practice, among others via the Dst architecture, the PP loss, and the temporal metrics. In both tasks, i.e. VSR and UVT, our results contain realistic spatial details that previous methods cannot produce, together with temporal coherence which is on par with state-of-the-art non-adversarial methods. The combination is crucial here, as non-adversarial methods can trivially achieve temporal coherency by generation outputs that lack detail.
>
> Q2: “I found the video generations impressive…although TecoGAN does not beat its competitors on most proposed metrics…”
>
> A2: Due to the complicated nature of the data, unfortunately, no approach currently exists that can reliably quantify the perceptual quality of natural images over time. Hence, we found it very important to consider spatial metrics (LPIPS, and PSNR as a very rough indicator) as a baseline for visual quality, and in addition, consider temporal metrics (tLP and tOF) to assess temporal changes. We have refrained from introducing a single criterion using a weighted combination of the different metrics, as this would raise fundamental questions about how to weight the individual terms.
>
> To give a specific example: compared with DUF, TecoGAN performs significantly better in terms of LPIPS, equally well on tLP&tOF, and only slightly worse in terms of PSNR. As reported by other works, the trade-off between PSNR and perceptual quality is unavoidable, therefore, when improving LPIPS from 2.607 (DUF) to 1.623 (TecoGAN), a PSNR decrease of less than 2dB is very reasonable. The TecoGAN model is on-par in terms of tLP&tOF despite the detail added by our method.
>
> To conclude, the metric data in our submission consistently shows that no other method achieves the same performance in terms of spatial detail and temporal coherence at the same time. This is confirmed by our user studies.
>
>
> Q3: “… I have concerns about the generalization of the approach…”
>
> A3: We agree that generalization is a crucial aspect for methods in this area. This is one of the motivations for evaluating our method thoroughly for two distinct application domains (most existing works target only one of the two). Within each application, we offer ablation studies, several video results, and metric evaluations. We performed user studies for VSR and we have additionally included new user studies for UVT (for details, cf. Q5 of Reviewer #4 above and “Rebuttal_R4.html”). For VSR, we train one TecoGAN model which performs very well for a wide range of video content. In UVT, we train one model for each dataset and always use the same code and parameters. In all ablation studies, the proposed temporal self-supervision offers clear and consistent improvements for a variety of VSR and UVT test cases (i.e., foliage and calendar scenes for VSR, Obama-and-Trump and smoke scenes for UVT). We believe that taken together, these results provide a detailed and varied assessment of the generality of our approach.
>
>
> Q4: “Is RecycleGAN unable to be applied to video super-resolution? Why is it not compared to in Table 2?”
>
> A4: The RecycleGAN algorithm is designed for UVT tasks. Besides a generator for single-image translation, they propose to use a predictor network to learn the temporal evolution within one video domain. While the predictor networks could be used instead of commonly used flow estimators for VSR, it would be very hard to get high-quality predictions because of the multi-modality of general video data. The generator network of RecycleGAN by itself does not stand out. Hence, an adoption of RecycleGAN for a task it was not designed for (VSR) is unlikely to behave better than state-of-the-art VSR methods like DUF and FRVSR, which is why we only compare RecycleGAN for the evaluation of UVT tasks.
>
>
> Q5: Is L_Phi the perceptual loss? I don't think it was mentioned in the body of the text.
> A5: Thanks, this is correct - we will add the corresponding explanation.

---

> ### Author Response · Authors · 2019-11-12
> **Reply to reviewer #1 (part 2)**
>
> Q6: I found the PP loss to be unjustified.
>
> A6: We found the PP loss to be crucial to tackle the issue of “streaking artifacts” being accumulated over time. This problem is a general one for recurrent models, as training sequences of arbitrary length is not possible with current hardware. In Fig 5 of our paper and the supp.-mat. webpage section 4.3, we show that the PP loss is very effective for avoiding this problem. The foliage scene has 40 frames. Our models are trained with sequences of length 10. The PP loss extends the length to 19. From videos, we see that without PP loss, strong artifacts become visible from frame 15 to 40. In contrast, the model trained with a PP loss is artifact free for all 40 frames.
>
> In the PP loss, there are two parts that contribute to solving the problem. One is the PP data augmentation and the other one is its temporal consistency constraint.
> 1) the PP data augmentation extends an N-frame sequence to the symmetric sequence of length (2N-1). This is helpful because many high-quality video data-sets only have few frames, e.g., the 3-frame and 7-frame Vimeo 90k datasets. With the PP data augmentation, we can train recurrent models to effectively see longer sequences.
> 2) the temporal constraint for the forward-backward pass. In order to show how much this aspect contributes separately, we carried out the following test.
>
> In addition to a regular model with both the PP augmentation (1) and the PP constraint (2), we trained another TecoGAN variant that only employs the PP data augmentation of (1) without the forward-backward pass constraint (i.e., λ_p=0 in the 4th row of Table 1). This model is denoted as “PP-Augment”. Video results can be seen in “Rebuttal_R1.html” contained in the “Rebuttal.zip” archive at: https://www.dropbox.com/sh/n07l8n51slh1e9c/AAAVngT9xsSzs1pJQqe5xV1Oa?dl=0
>
> So during training, the generator network of DsDt sees 10 frames, while the generators of PP-Augment and TecoGAN{-} see 19 frames. While DsDt shows strong recurrent accumulation artifacts early on (from frame 15), the PP-Augment version slightly reduces the artifacts. It works good for frame 15 but shows artifacts from frame 32 on. Only our regular model (TecoGAN{-}) successfully avoids temporal accumulation for all 40 frames. With the PP constraint (2), the model can avoid the recurrent accumulation of artifacts and work well for sequences that are substantially longer than the training length. Among others, we have tested our model with ToS sequences of lengths 150, 166 and 233. For all of these sequences, TecoGAN model did not show temporal accumulation or streaking artifacts.
>
> Thus, the PP loss not only provides augmentation, but the temporal constraint is crucial for avoiding visual artifacts. We will include this new comparison in the revised version of our document.

---

### Official Review · AnonReviewer2 · 2019-10-24
**Official Blind Review #2**

**Rating:** 8

**Review:**


The paper presents a novel method for training video-to-video translation (vid2vid) models. The authors introduce a spatio-temporal adversarial discriminator for GAN training, that shows significant benefits over prior methods, in particular, parallel (as opposed to joint) spatial and temporal discriminators. In addition the authors introduce a self-supervised objective based on cycle dependency that is crucial for producing temporally consistent videos. A new set of metrics is introduced to validate the claims of the authors.

I really like this paper. Although the method is a rather complex mix of multiple losses, these are justified in detail, both intuitively and empirically. The appendix is filled with much more detail about implementation, architecture and more results. Finally the results show that the proposed method is superior across the board compared to previous approaches from the literature.


Strenghts:
- The approach works on two distinct applications of vid2vid.
- Detailed ablations justifying the introduction of every single part of the overall objective.
- Strong results.
- Clear writing and presentation.
- A lot of additional details and results in the appendix for the more interested reader.

Weaknesses:
- Rather complex overall objective
- Seems to need a lot of tweaking

Questions:
- Isn't the PP loss just an incarnation of cycle consistency loss? If so, maybe there is no need for the introduction of a new name for it.


**Experience Assessment:**

I do not know much about this area.

**Review Assessment: Checking Correctness Of Derivations And Theory:**

I assessed the sensibility of the derivations and theory.

**Review Assessment: Checking Correctness Of Experiments:**

I assessed the sensibility of the experiments.

**Review Assessment: Thoroughness In Paper Reading:**

I read the paper at least twice and used my best judgement in assessing the paper.

---

> ### Author Response · Authors · 2019-11-12
> **Reply to reviewer #2**
>
>
> Dear reviewer, thank you very much for the review and comments. We are glad to hear the positive assessment of our results.
>
> Q1: “Rather complex overall objective”
>
> A1: We agree that our learning objectives look complex because of the two distinct application domains. We found it important to show as much as possible that our approach is of general interest for difficult spatio-temporal data. We will try to further clarify the differences and similarities between VSR and UVT formulations in future revisions of our text.
>
>
> Q2: “Seems to need a lot of tweaking”
>
> A2: GANs are generally non-trivial to train, and our models share this behavior. In practice, we found working with the proposed architectures not to be overly difficult. We recommend starting with a stable version for spatial single-image inference, and then extending it with our main contributions for temporal self-supervision. For the VSR and UVT tasks, we found that once a balanced spatial GAN (single-image SR or single image translation) is realized, the temporal extension, i.e., adding a frame-recurrent input to the generator and changing the Ds to a Dst architecture, did not require further hyper-parameter fine-tuning.
>
> In addition, two hyper-parameters, one to train the flow estimator F and one for the PP loss, need to be adjusted. These hyper-parameters can be chosen based on shorter test runs. In practice, we write results for a fixed set of validation samples to disk. Based on the evolution of these samples over the course of training, especially with respect to spatial detail and temporal stability, suitable parameters can typically be chosen after a small number of test runs.
>
>
> Q3: “Isn't the PP loss just an incarnation of cycle consistency loss?”
>
> A3: The “cycle-consistency” loss refers to different constraints in different settings. While some of them share similar ideas with our PP loss at a high level, we consider the proposed PP loss to be novel and especially useful for generative tasks in the area of videos. Our formulation is tailored to the adversarial learning of detailed content, and it is key for preventing artifacts from being accumulated over time for recurrent models.
>
> Considering specific prior work, CycleGAN proposes the spatial cycle-consistency loss while RecycleGAN proposes a cycle loss across data domains and time. The PP loss instead aims for a strict temporal coherence constraint and is inherently different from both.
>
> Beyond the field of video generation, the papers on time cycle loss used for optical flow estimation and tracking is closer to the PP loss at a high level. In these areas, the goal is to infer valid motions or tracking over frames, and a similar L2 loss is used to constrain a fixed video warped with the inferred motions. This time cycle loss is used to optimize indirectly via the estimated motion/tracked positions. It can then improve the accuracy of the tracked positions or estimated motions. We instead use the PP loss to directly constrain the generated video content. In this setting, the PP loss successfully improves the long-term temporal consistency of the results. Due to the different goals of our PP loss and the concurrent nature of the other work in the area, we kept the different name of our formulation.

---

### Official Review · AnonReviewer3 · 2019-11-03
**Official Blind Review #3**

**Rating:** 6

**Review:**

The paper presents video generation method with spacio-temporally consistent features. This is done through: a) temporal adversarial learning, b) Ping Pong loss, and c) metrics that quantify the quality. The methods are evaluated on two datasets and user studies.

The idea is interesting and the paper is well written. The results are convincing.

The originality of the concatenation of several frames is somewhat limited, since it is a standard procedure in other domains such as robotics. Nevertheless the results are positive.

Seems like the metrics definitions were not included in the main body of the paper - the authors should either include them to remove from the contributions.

**Experience Assessment:**

I do not know much about this area.

**Review Assessment: Checking Correctness Of Derivations And Theory:**

I assessed the sensibility of the derivations and theory.

**Review Assessment: Checking Correctness Of Experiments:**

I assessed the sensibility of the experiments.

**Review Assessment: Thoroughness In Paper Reading:**

I read the paper at least twice and used my best judgement in assessing the paper.

---

> ### Author Response · Authors · 2019-11-12
> **Reply to reviewer #3**
>
> Dear reviewer, thank you very much for the review and suggestions.
>
> Q1: “originality of the concatenation of several frames is somewhat limited”
> A1: It is correct that such a concatenation is used in a variety of other settings. The core aim of our work is to highlight its importance in a very challenging setting, i.e., for training spatio-temporal GANs for natural image sequences. In this field, L2-based losses dominate, and with our work, we demonstrate how much additional detail can be generated with the right approach for adversarial training. We will clarify this in the text of our submission.
>
> Q2: “the metrics definitions were not included in the main body of the paper”
> A2: We will move the metrics definitions into the main part in a future version of the paper.

---

### Official Review · AnonReviewer4 · 2019-11-04
**Official Blind Review #4**

**Rating:** 3

**Review:**


Summary:
This paper proposes a training objective for higher quality video generation for the tasks of Video Super Resolution (VSR) and Unpaired Video Translation (UVT) and also two evaluation metrics tOF and tLP. They provide a comprehensive ablative study of the proposed method and also show comparisons against baselines for the tasks of VSR and UVT.


Pros:
+ Novel video generation method for VSR and UVT.
+ Novel metrics
+ Well written paper

Weaknesses / comments:

- Confused about inputs to discriminator (I_{w,g} and I_{w,b})
I am not sure I understand the inputs I_{w,g} and I_{w,b} to the discriminator network. Based on the description, I_{w,g} = {W(g_{t-1}, v_t), g_t, W(g_{t+1}, v_t^’)} = {g_t, g_t, g_t}, assuming v_t^” means reverse flow. A similar process seems to be happening with I_{w,b}. If this is the case, will the discriminator optimally get the same frame concatenated together (i.e. {g_t, g_t, g_t})? Now, if that’s the case, how is the discriminator modeling temporal consistency other than making the generator learn to put pixels forward and back in place since the “original triplets” (real data) are {g_{t-1}, g_t, g_{t+1}} and {b_{t-1}, g_t, g_{t+1}}, respectively. This is very confusing to me. It would be good if the authors can clarify this in the rebuttal.


- How do you prevent the zero output in PP loss?
The PP-loss compares forward and backward prediction by || g_t - g_t^’ ||. If not careful, the neural network can just learn to output zeros or the same frame for the full video. Did the authors see any behavior like this? Or did the proposed formulations prevent this? Or was there a very small weight applied to this loss?



- TecoGAN vs baselines generator parameters (rather than TecoGAN^{-}).
Based on the experimental section, it looks like TecoGAN is the main network being compared to the baseline methods. TecoGAN has more parameters in the generator compared to TecoGAN^{-}. Did the authors make sure that the generator had the same number of parameters as the other methods? The authors mention a performance difference between TecoGAN and DUF due to DUF having more parameters, but what about the other baselines?

- Evaluation metric contributions in the supplementary material?
The description of the two proposed metrics tOF and tLP have been placed in Appendix B. These are mentioned as contributions of the paper so their description should be in the main text. Please make sure that Appendix B is in the main text in future versions of this paper.


- UVT task evaluation in the supplementary material?
The UVT task is mentioned in the abstract and as a target task in this work, however, the evaluations for this are in the Appendix. Please move them to the main text in future versions of this paper. Secondary experiments should be in the Appendix but I feel this is primary.


- UVT task only evaluated with the proposed evaluations?
The UVT evaluation in the Appendix only has the proposed metrics as evaluation. I understand that, since it’s an unpaired video translation task, there is no ground truth to compare against. However, a human based study could be done where human raters would judge for the more realistic of N videos.


- The same qualitative comparisons in the paper are not in the provided website.
The provided website does not have the same qualitative evaluations as the paper does. For example: There are 7 methods being compared for the same video in Figure 8 in the supplementary material, but I don’t see anything like that in the website.


- For the UVT task, we omit the DsDtPP model because …….
In the paper, the authors mention that they don’t provide the DsDtPP baselines because it requires a lot of computation. However, I feel these missing baselines make the experiments incomplete since it’s a different task.


- Figure 5 has no point of reference with which to compare the frames.
Figure 5 points out that the video b) is better than video a). While it is true that one seems more noisy than the other, there should be a point of reference for readers to make sure of it.


Conclusion:
In conclusion, the paper seems to present a novel method and evaluation metrics but has many issues as stated above. It would make the submission better if the authors can address them in the rebuttal.

**Experience Assessment:**

I have published in this field for several years.

**Review Assessment: Checking Correctness Of Derivations And Theory:**

N/A

**Review Assessment: Checking Correctness Of Experiments:**

I carefully checked the experiments.

**Review Assessment: Thoroughness In Paper Reading:**

I read the paper thoroughly.

---

> ### Author Response · Authors · 2019-11-12
> **Reply to reviewer #4 (part 1)**
>
> Dear reviewer, thank you very much for the detailed review and comments. We are glad to hear that you consider the paper to be well written and the method to be novel. The comments about unclear parts and potential weaknesses are very important for us to refine and improve our paper. Below, we will address your questions in more detail.
>
> Please note that for questions Q1, Q5 and Q8 additional examples/results were required. We provide these via the file “Rebuttal_R4.html” contained in the “Rebuttal.zip” archive at: https://www.dropbox.com/sh/n07l8n51slh1e9c/AAAVngT9xsSzs1pJQqe5xV1Oa?dl=0
>
>
> Q1: “Confused about inputs to discriminator (I_{w,g} and I_{w,b})“
>
> A1: When the generated results I_{g} are artifact-free (i.e. do not jitter), it is correct that the triplet I_{w,g} warped with the forward and reverse motions (v_t and v_t') would simply contain three copies of the middle frame. The same holds for the ground-truth triplet I_{w,b}. However, in practice, the motions (v_t and v_t’) cannot capture evolutions such as changing illumination, or occlusions. In addition, the motion fields contain approximation errors. Our tests show that these residual changes in the inputs are what allows the discriminator to learn about natural motions. (Specific examples are shown in the Rebuttal_R4.html page.)
>
> Intuitively, the warped frames make the job easier for the discriminator by supplying information “in-place” wherever possible, such that it can assess how natural the changes are. At the same time - as the motions are potentially imperfect - we found that the discriminator benefits from being able to fall back to the original (unwarped) triplets.
>
> Our ablation studies (Supp.-mat. webpage section 4.2, and the third column of section 5), show that adding the warped frames consistently and reliably improves the temporal behavior. D_{st} can learn complex temporal functions by supervising via warped triplets and original triplets, and thus provides good learning objectives for the generator. In practice, we have not encountered situations where the generator learns to simply map pixels or structures backward and forward. It is more natural for the generator to rely on the provided input data, i.e. the previous generated frame and the current low-resolution frame, to infer outputs.
>
>
> Q2: “How do you prevent the zero output in the PP loss?”
>
> A2: Being one of the possible minima, a zero output trivially satisfies the PP loss formulation, but is easily identified by the discriminator D_st. Hence, such an output typically yields a large adversarial loss. Considering one pixel (i,j) in a forward sequence g_t(i,j), the PP loss also considers a pixel from the backward pass g_t’(i,j). Assuming that g_t(i,j) contains a smaller intensity than g_t’(i,j), the gradient of the PP loss will lead to an increase of g_t, while g_t’ will be decreased. At the same time, the gradient of D_{st} will require g_t and g_t’ to match the reference distribution, i.e., to contain natural and sharp details. So the PP loss only requires g_t and g_t’ to be the same. An L2 distance is a right choice here as g_t and g_t’ are perfectly aligned, and the L2 distance can be correctly minimized even for detailed content. There is no warping or occlusion, and the data by construction is not multi-modal.
>
> Taking the VSR task as an example, the hyper-parameter for the PP loss is 0.5. At the end of the training, the total generator loss of the TecoGAN model has 76.4% adversarial loss, 6.8% PP loss and 16.8% other losses (averaged across 4000 frames). Hence, the adversarial component is dominating, but the PP loss still yields a substantial contribution. When we instead evaluate the PP loss for the DsDt model in the same way (it was not used for training the DsDt model), the loss is 49.1% higher than for the TecoGAN model. So the PP loss is successfully and substantially reduced without converging to a trivial zero solution.

---

> ### Author Response · Authors · 2019-11-12
> **Reply to reviewer #4 (part 2)**
>
> Q3: “TecoGAN vs baselines generator parameters (rather than TecoGAN^{-}).”
>
> A3: Model Sizes: the following table summarizes the sizes of the different models compared in our evaluation section:
>
>     Model                  Weight Count
> +-------------------+-----------------------------------------------------------+
> | DUF                | 6.2 million —— 105% more than TecoGAN |
> +-------------------+-----------------------------------------------------------+
> | FRVSR             | 0.84 million for super-resolution,                 |
> |                         | 1.74 million for flow estimation.                   |
> +-------------------+-----------------------------------------------------------+
> | ENet                | 0.84 million —— single-image SR                 |
> +--------------------+-----------------------------------------------------------+
> | TecoGAN^{-}  | 0.84 million for the generator,                       |
> |                          | 1.74 million for flow estimation                     |
> |                          | —— same as FRVSR                                         |
> +--------------------+-----------------------------------------------------------+
> | TecoGAN         | 1.29 million for the generator,                       |
> |                          | 1.74 million for flow estimation                      |
> |                          | —— 17% more than FRVSR or TecoGAN^{-} |
> +--------------------+------------------------------------------------------------+
>
> - TecoGAN vs. FRVSR, DUF, ENet:
> To summarize, TecoGAN has 17% more weights than FRVSR and DUF has 105% more weights than TecoGAN. Although the weight count is different for each model, the fundamental difference between them are the learning objectives: Even if we increase the size of FRVSR, it will still generate results that lack detail because of the averaging nature of its L2 loss.
>
> - TecoGAN{-} vs. FRVSR:
> For the Vid4 dataset, we show all the metrics in Table 2. Comparing TecoGAN^{-} and FRVSR, with the same size, TecoGAN^{-} achieves a much better LPIPS score, slightly better scores in terms of tOF, tLP, and a slightly worse PSNR score. Fig. 6 highlights that TecoGAN^{-} generates substantially more detail than FRVSR. We also provide the outputs of TecoGAN^{-} now for Vid4 (see Q6).
>
>
> Q4: “Evaluation metric contributions in the supplementary material?” & “UVT task evaluation in the supplementary material?”
> A4: Thank you for the suggestion. We’re glad to hear the metrics and UVT evaluation are considered to be important, we will include them in the main document of our submission.
>
> Q5: “UVT task only evaluated with the proposed evaluations? ...a human-based study could be done…”
>
> A5: It is a good suggestion to evaluate the UVT results with user studies. As no ground-truth data is available, we carried out two sets of user studies:  One uses an arbitrary sample from the target domain as the reference and the other uses the actual input from the source domain as the reference. Comparing CycleGAN, RecycleGAN and TecoGAN on the Obama-and-Trump data-set, we found that our TecoGAN results are clearly preferred in both cases. Details of these results can be found in the Rebuttal_R4.html page, and we will include the results of these user studies in future revisions of the paper.
>
>
> Q6: The same qualitative comparisons in the paper are not in the provided website.
> A6: We did not upload these videos as part of our submission due to the relatively large amount of material in the appendix and supplemental webpage. We are happy to provide additional comparison videos for the different methods.
>
> The following sequences can now be found in the VSR_videos.html page of the “Vid4-ToS-Videos.zip” archive at https://www.dropbox.com/sh/n07l8n51slh1e9c/AAAVngT9xsSzs1pJQqe5xV1Oa?dl=0:
>   - the ground-truth and results for 8 methods on the Vid4 dataset
>   - the GT and results for 4 methods on the ToS dataset and lizard, armor, spider scenes.
>
> These sequences are the originals that were used to extract the single frames shown in our submission. As visible in the static figures in the PDF, adversarial learning yields more details across the different scenes. The videos highlight that our TecoGAN model additionally yields excellent coherence over time.

---

> ### Author Response · Authors · 2019-11-12
> **Reply to reviewer #4 (part 3)**
>
> Q7: “For the UVT task the DsDtPP model was omitted”
>
> A7: We have included an additional DsDtPP model for comparison. As we mentioned in the main text, it is necessary to find a good balance between the two generators and the four discriminator networks. By weighting the temporal adversarial losses from Dt with 0.6 and the spatial ones from Ds with 1.0, the DsDtPP model yields similar performance to the Dst model on the smoke dataset. A result generated with this model can be found on the Rebuttal_R4.html page. We can include this model in our submission for completeness, but we would like to point out that the proposed Dst architecture is the better choice in practice, as it learns a natural balance of temporal and spatial components by itself and requires fewer resources.
>
>
> Q8: Figure 5 has no point of reference with which to compare the frames.
> A8: We will include a ground truth image in our revision for comparison.
>
> Thanks again.

---

### Author Response · Authors · 2019-11-12
**Additional materials required/suggested by reviewers**


We would like to thank all the reviewers for their insightful comments and suggestions. In order to address the questions raised in the reviews, we have uploaded additional materials to shed light on the behavior and performance of our approach.

The corresponding experiments can be seen via the dropbox link: https://www.dropbox.com/sh/n07l8n51slh1e9c/AAAVngT9xsSzs1pJQqe5xV1Oa?dl=0
Please download the “Rebuttal.zip” file, and open the Rebuttal_R*.html page contained in it.

Most importantly, we show:
- A series of new user studies to evaluate the unpaired video translation task. (Rebuttal_R4.html)
- Example triplets to illustrate the importance of warped and unwarped content (Rebuttal_R4.html)
- Outputs of a UVT DsDtPP model for comparison (Rebuttal_R4.html)
- An ablation study for the data-augmentation versus temporal constraint properties of the proposed PP loss. (Rebuttal_R1.html)

Additionally, all VSR videos shown in our paper can be found on the "VSR_videos.html" webpage inside the "Vid4-ToS-Videos.zip" archive. A different file is created due to the large file size (196MB).

---

> ### Author Response · Authors · 2019-11-14
> **an updated draft with modifications highlighted**
>
> We updated our draft according to the suggestions.
> New parts are highlighted in the document.
> We thank the reviewers for their help to improve the document.

---

### Decision · Program_Chairs · 2019-12-19

**Decision:**

Reject

**Comment:**

The paper presents an architecture for conditional video generation tasks with temporal self-supervision and temporal adversarial learning. The proposed architecture is reasonable but looks somewhat complicated. In terms of technical novelty, the so-called "ping-pong" loss looks interesting and novel, but other parts are more-or-less some combinations of existing techniques. Experimental results show promise of the proposed method against selected baselines for video super-resolution (VSR) and unpaired video-to-video translation tasks (UVT). In terms of weakness, (1) the technical novelty is not very high; (2) the final loss is a combination of many losses with many hyperparameters; (3) experimentally the proposed method is not compared against recent SOTA methods on VSR and UVT.

The proposed method should be compared against more recent SOTA baselines for VSR tasks (see examples of references below):

EDVR: Video Restoration with Enhanced Deformable Convolutional Networks
https://arxiv.org/abs/1905.02716

Progressive Fusion Video Super-Resolution Network via Exploiting Non-Local Spatio-Temporal Correlations
ICCV 2019

Recurrent Back-Projection Network for Video Super-Resolution
CVPR 2019

The same comment would apply for baselines for UVT tasks:

Mocycle-GAN: Unpaired Video-to-Video Translation
https://arxiv.org/abs/1908.09514

Preserving Semantic and Temporal Consistency for Unpaired Video-to-Video Translation
https://arxiv.org/abs/1908.07683

Particularly for UVT, the evaluated dataset seems limited in terms of scope as well (i.e., evaluations on more popular benchmarks, such as Viper would be needed for further validation). Overall, given that the contribution of this work is an empirical performance with a rather complex architecture/loss, more comprehensive empirical evaluations on SOTA baselines are warranted.